# The Morphology of Ice and Liquid Brine in the Environmental SEM: A Study of the Freezing Methods

Ľubica Vetráková,[1] Vilém Neděla,[1] Jiří Runštuk,[1] Dominik Heger[2]

[1]Environmental Electron Microscopy Group, Institute of Scientific Instruments of the Czech Academy of Sciences, Brno, Czech Republic

[2]Department of Chemistry, Faculty of Science, Masaryk University, Brno, Czech Republic

Correspondence to: Ľubica Vetráková (vetrakova@isibrno.cz), Dominik Heger (hegerd@chemi.muni.cz)

## Abstract

The microstructure of polycrystalline ice with a threading solution of brine controls its numerous characteristics, including the ice mechanical properties, ice-atmosphere interactions, sea-ice albedo, and (photo)chemical behavior in/on the ice. Ice samples were previously prepared in laboratories to study various facets of ice-impurities interactions and (photo)reactions to model natural ice-impurities behavior. We examine the impact of the freezing conditions and solute (CsCl used as a proxy for naturally occurring salts) concentrations on the microscopic structure of ice samples via an environmental scanning electron microscope. The method allows us to observe in detail the ice surfaces, namely, the free ice, brine puddles, brine-containing grain boundary grooves, individual ice crystals, and imprints left by entrapped air bubbles at temperatures higher than −25°C. The amount of brine on the external surface is found proportional to the solute concentration and is strongly dependent on the sample preparation method. Time-lapse images in the condition of slight sublimation reveal sub-surface association of air bubbles with brine. With rising temperature (up to −14 °C), the brine surface coverage increases to remain enhanced during the subsequent cooling and until the final crystallization below the eutectic temperature. The ice recrystallization dynamics identify the role of surface spikes in retarding the ice boundaries' propagation (Zeener pining). The findings thus quantify the amounts of brine exposed to incoming radiation, available for the gas exchange, and influencing other mechanical and optical properties of ice. The results have straightforward and indirect implications for artificially prepared and naturally occurring salty ices, respectively.

## 1  Introduction

Ice and snow are important reaction media in which chemical compounds (impurities) can be accumulated, transformed, and released back to other compartments in the environment (Bartels-Rausch et al., 2014). The location and time evolution of impurities within a frozen sample are crucial in multiple respects: they determine the mechanical and optical properties of the material and thus must be considered in relation to snowpack stability, avalanches, sea ice mechanics, and climate change research (Blackford et al., 2007;Hobbs, 2010;Dash et al., 2006;Wåhlin et al., 2014). The location of impurities most probably influences the compounds' reactivity because their availability to incoming light radiation and/or gaseous reactants substantially differs depending on whether the impurities are located on the ice surface or buried in the frozen bulk (Hullar and Anastasio, 2016;Bartels-Rausch et al., 2014). However, the information combining the reactivity of compounds with their locations is essentially missing. Excepting low concentrations of HF, $NH_4^+$, HCl, $HNO_3$ and formaldehyde (Perrier et al., 2002;Thibert and Domine, 1998, 1997), impurities are usually not incorporated into the ice lattice (Krausková et al., 2016;Hobbs, 2010;Wilson and Haymet, 2008). As ice is highly intolerant to impurities, these are segregated to an unfrozen freeze-concentrated solution (FCS) during the growth of ice crystals. The FCS can be located in the lamellae and veins between the ice crystals; (micro)pockets

within the ice structure; or puddles and grain boundary grooves on the ice surface. When the mixture is cooled below the eutectic temperature ($T_{eu}$), the FCS may crystallize or vitrify (Salnikova et al., 2015;Bogdan et al., 2014).

To determine the actual positions of chemical compounds in ice samples, various microscopic techniques are used, including optical (Bogdan et al., 2014;Bogdan and Molina, 2017), fluorescence (Cheng et al., 2010;Roessl et al., 2015), electron (Blackford et al., 2007;Chen and Baker, 2010;Rosenthal et al., 2007;Barnes et al., 2002), and confocal Raman (Dong et al., 2009) microscopies. While optical microscopy can be applied at environmentally relevant temperatures and pressures, very low pressure conditions ($10^{-4}$ to $10^{-2}$ Pa) are needed in the specimen chamber of a conventional scanning electron microscope (SEM). At such low pressures, ice sublimes very rapidly due to its high vapor pressure (Blackford, 2007); for that reason, ice samples are observable only at very low temperatures (below $-120$ °C) in a SEM. (Blackford et al., 2007) These conditions are far from naturally relevant ones. Conversely, however, electron microscopy resolves fine structures two orders of magnitude smaller compared to those examined with optical microscopy, and its depth of field is much larger. Moreover, the ice surface appears opaque in a SEM, and its surface topography is thus defined significantly better than possible via optical microscopy (Blackford, 2007). In order for electron microscopy to be usable at higher pressures, the environmental scanning electron microscope (ESEM) was designed (Danilatos, 1993). The ESEM can operate at specimen chamber pressures of up to thousands of Pascals, and such capability allows frozen samples to be inspected at higher temperatures, even up to the samples' melting point. To prevent ice sublimation, water vapor can be purged into the specimen chamber; thus, the imaging is performed under closer-to-natural conditions. Further, ionization of the gas in the specimen chamber of the ESEM prevents the sample from being charged, rendering the technique very convenient for imaging electrically non-conductive samples without the need of conductive coating. The ESEM is therefore becoming increasingly popular within research into pure ice (Nair et al., 2018;Magee et al., 2014;Chen et al., 2017) and ice-impurities interactions under static and dynamically changing conditions (Krausko et al., 2014;Yang et al., 2017). While electron microscopy visualizes the sample surface in detail, micro-computed tomography was recently applied to investigate solute locations in the frozen bulk (Hullar and Anastasio, 2016). Using this technique, the authors were able to visualize the locations of solutes (CsCl or rose bengal), gas, and ice in 3D.

Salt solutions are abundant in the natural environment; seawater alone represents 96.5% of water on the Earth (Gleick, 1993). In wintertime, sea ice covers an area of up to 7% of the Earth's surface and, as such, embodies one of the largest biomes (Thomas, 2017). Salts and ice also coexist in sea aerosols (Knipping et al., 2000;Zobrist et al., 2008) and in coastal regions, on frost flowers (Douglas et al., 2012),  and also in ice cores (Ohno et al., 2006, 2005;Vega et al., 2018).

Well-conceived laboratory investigation of the ice-impurity interaction has the potential to simplify complex natural systems. Laboratory-based studies of the (photo)chemical reactions of frozen aqueous solutions have improved the understanding of the (photo)chemistry occurring in cold environments (Bartels-Rausch et al., 2014;Klanova et al., 2003). However, the dissonance between results from individual laboratories as regards the photodegradation rates (Kahan et al., 2010;Ram and Anastasio, 2009) points to the fact that there still remain key factors influencing the reactivity in a frozen solution which were not comprehended previously.

The impurities' locations were most often deduced from the sample preparation methods; for example, freezing an aqueous solution involved the precondition of placing the impurities inside the ice matrix. Subsequent breaking of the ice into small pieces was proposed to bring the impurities on the ice surface (Kahan et al., 2010); this assumption was recently questioned (Hullar et al., 2018). Other studies suggested that freezing an aqueous solution places the impurities prevalently inside the ice interior, whereas deposition of the organics from the vapour phase accommodates them on the ice surface (Ondrušková et al., 2018;Vetráková et al., 2017;Krausko et al., 2015b;Krausko et al.,

2015a;Kania et al., 2014;Heger et al., 2011;Hullar et al., 2018;Bartels-Rausch et al., 2013). Conversely, nebulizing the solution into liquid nitrogen was assumed to produce ice spheres with the organic impurities on their surfaces (Kurkova et al., 2011).

To establish a direct proof of salt location in frozen samples, we herein applied various methods to freeze caesium chloride (CsCl) solutions. We had intended to determine how the ice impurities' location and surface availability are affected by the freezing technique, temperature, and salt concentrations. The ESEM technology was utilized for the visualization due to its ability to represent the samples' surface topography together with the impurities' locations. CsCl was chosen as the model salt, being similar to NaCl and providing very good microscopic contrast; thus, the presence of brine on the surface of the frozen samples can be clearly monitored, with indirect implications towards the location of sea water brine on sea ice.

## 2   Methods

The microscopic images were recorded using an AQUASEM II, a Tescan VEGA SEM modified at the Institute of Scientific Instruments of the Czech Academy of Sciences (Tihlarikova et al., 2013;Nedela, 2007). This ESEM is capable of imaging wet, electrically non-conducting samples at chamber pressures as high as 2,000 Pa and temperatures of down to −27 °C. The advantages of the AQUASEM II in the imaging of frozen samples had been described previously (Krausko et al., 2014;Yang et al., 2017).

### 2.1 Preparation of the samples

CsCl solutions with the concentrations of 0.005, 0.05, and 0.5 M were prepared by dissolving an appropriate amount of CsCl in MilliQ $H_2O$. The molar concentrations correspond to molalities of 0.005, 0.050, and 0.512 mol/kg, respectively, using the solution densities at 20 °C published by Reiser et al. (2014); the positions of these concentrations can be realized on the phase diagram for the CsCl-water (Figure 1) (Gao et al., 2017). The concentrations are proxies for natural salt occurrence in coastal snow (5 mM) (Beine et al., 2012;Douglas et al., 2012), up to the concentration in sea water (0.5 M) (Massom et al., 2001;Thomas, 2017). Prior to the measurement, the solutions had been filtered through a 0.45 μm filter to exclude impurities that might interfere with the microscopic observations. The solutions were frozen at atmospheric pressure via four distinct methods:

(I)     Spontaneous freezing of a droplet without seeding. A droplet was put onto the silicon sample holder of the ESEM at room temperature (about 23 °C). The temperature of the sample holder, controlled by a Peltier stage, was lowered gradually (the cooling rate corresponded to approximately 0.5 °C/s) until the sample froze spontaneously, with the estimated freezing rate being (154 ± 13) mm/s. The nucleation temperature ranged from −16 to −18 °C in the replicate experiments. After becoming frozen, the sample was cooled down to the initial observation temperature of −22 to −25 °C.

(II)    Freezing of a droplet with seeding. A droplet was placed onto the sample holder of the ESEM, and its temperature was set to −2 °C; then, after thermal equilibration, several small ice crystals were added to the edge of the sample to initiate the nucleation process. The estimated freezing rate reached about 0.2 mm/s. Subsequently, the temperature of the sample holder was lowered down to the initial observation value.

(III)   Spraying into a vessel containing liquid nitrogen (LN). Small ice spheres (similar to artificial snow) were formed and then transferred onto the microscope sample holder precooled to −25 °C.

(IV)    Fragmentation of an LN-frozen sample. The applied CsCl solution enclosed in a hard gelatine capsule (diameter of 8 mm; approx. height of 25 mm, the sample level was at ~10 mm) was immersed in LN. After freezing, the sample was fragmentized with a scalpel; thus, irregularly

shaped pieces were formed, and the former interior of the sample was revealed. The fragments from the interior of the sample were transferred onto the ESEM sample holder precooled to −25 °C.

In the methods I and II, the freezing rates were estimated by observing the process of freezing the 0.05 M CsCl solution with a high-speed camcorder (Phantom V611, Vision Research, Inc., 3000 fps) and via analyzing the movement of the freezing interface. As regards the method I, the interface was nicely visible, and the calculated freezing rates were well reproducible; the method II, however, exhibited a very poorly visible interface, allowing only rough estimation of the freezing rate.

During the open-chamber freezing process, a thin layer of ice from the condensed moisture was formed on the surfaces of the frozen samples in all of the preparation methods. This layer protected the samples from sublimating while the specimen chamber was being pumped down. The layer exhibited a structure very different from that the frozen samples (Figure S18); thus, the desublimed ice was always readily distinguishable from the original ice sample. The sublimation of the condensed layer was monitored at the start of the ESEM imaging; as soon as the layer had sublimed, the imaged surface of the frozen sample was affected by the sublimation process to only a very small extent. We can infer this fact because we had also followed further sublimation of the ice samples, as will be reported in the near future. By further extension, the effect of ice sublimation could be more pronounced in temperature cycling and ice dynamics experiments.

| | The temperature of the cooling medium* | Sample size | Observed part |
|---|---|---|---|
| I | −16 °C | $d \approx 4$ mm<br>$h \approx 1$ mm | surface |
| II | −2 °C | | surface |
| III | −196 °C | $d \approx 100\text{-}300$ μm | interior |
| IV | | $d = 8$ mm<br>$h \approx 10$ mm | interior |

**Table 1.** The major parameters of the freezing methods I-IV. *The temperature of the cooling medium just before the nucleation started.

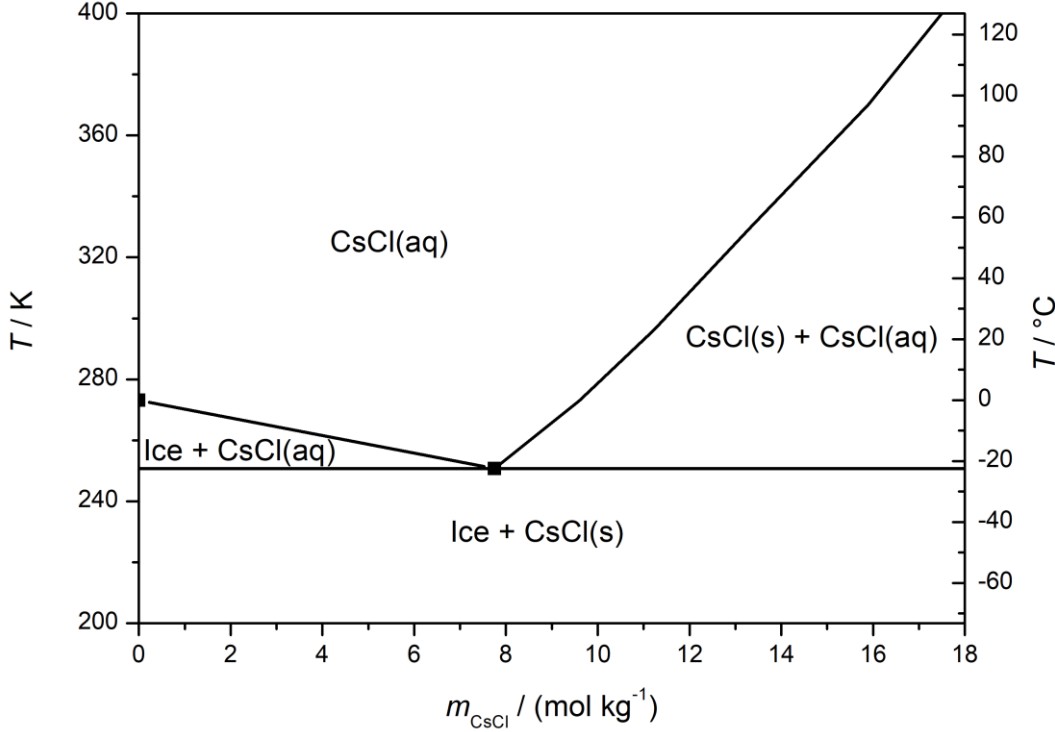

**Figure 1.** The CsCl-water phase diagram; adopted from Gao et al. (2017).

## 2.2 Imaging in the ESEM

The electron beam energy of 20 keV and a YAG:Ce[3+] scintillation backscattered electrons (BSE) detector, sensitive to the region encompassing the top 1,500 nm, were employed for the imaging. In a number of instances, the samples were represented with an ionization detector of secondary electrons, recording mainly signals from the thin subsurface layer (up to 5 nm); the device enabled us to obtain information about the topology of the surface (Neděla et al., 2015). In all of the measurements, a low beam current (40 pA) and a short dwell time (14 μs) were employed to minimize radiation damage to the frozen samples. During the experiments, we maintained the nitrogen gas partial pressure of 500 Pa inside the specimen chamber. Besides the nitrogen gas, water vapor was also purged into the chamber; the relative humidity was kept slightly below the sublimation curve values to prevent moisture condensation on the ice surface. Somewhat slow ice sublimation was observed, as conditions exactly matching the sublimation curve are difficult to establish inside the specimen chamber of the ESEM. For a fresh sample, the initial temperature of the sample holder was usually set to −25 °C (occasionally being −22 or −23.5 °C). Then, the temperature was gradually changed, and its impact on the brine abundance on the surface of the sample was monitored.

The indicated temperatures relate to the Peltier stage placed inside the sample holder. However, the temperature of the sample differs from that of the Peltier stage; we detected differences of up to 2 °C between the data provided by the sensor of the stage and the temperature of the sample measured with both a sensor frozen inside the ice and a thermal camera (Flir A310) in atmospheric conditions. The temperature settled significantly more slowly in the sample than in the Peltier stage: We observed a lag of about 3 minutes when the value of -18 °C was to be reached. Additionally, heating the sample surface with an electron beam (considering also the relatively warm gas purged through the specimen chamber) and the subsequent cooling due to the ice sublimation and/or water evaporation from the brine might affect the surface temperature during the imaging. The images were

typically recorded with the magnification of 500 (image resolution of ～ 0.5 µm), although a resolution
up to ten times higher is feasible.

## 2.3 Estimating the brine surface coverage

We used a BSE detector having high sensitivity to atoms with large atomic numbers; thus, in the
grayscale ESEM images, the sample surface covered with the CsCl is white, whereas the gray areas
represent the ice. To estimate the extent of the sample surface covered with the brine, Mountains®
software (Digital Surf) was applied. This software facilitates selecting pixels with brightness above the
chosen threshold; the pixels are then highlighted. By overlaying the highlighted mask and the white
areas in the original image, the best-fitting threshold is chosen to represent the brine and to exclude
the ice (Figure S1).
A constant threshold could not be employed for all of the images due to the fact that, during the
actual image acquisition, the parameters of the detector (and thus the brightness and contrast of the
image) were often changed based on the intensity of the detected signal to avoid under- or
overexposure. If the signal is locally too intensive (e.g., there are spots with a large amount of the
brine), the detector sensitivity must be reduced during the image acquisition to prevent
oversaturation; thus, spots with a low amount of the brine might not be visible in the image, and the
surface coverage could become underestimated. Conversely, with the CsCl signal too low, the
brightness of the brine might be comparable to that of the surface irregularities (as the detector is also
sensitive to the morphology of the sample, albeit to a limited extent only), leading to them being
misinterpreted as small brine-containing spots. Such a situation would then cause overestimated brine
surface coverage. The estimated coverage can therefore be directly compared exclusively in samples
with similar amounts of the salt (similar concentrations), where the detector response differences are
negligible. The uncertainty of the manual threshold selection procedure is indirectly deducible from
the variance of the surface coverage in the sequence of the images recorded at the same temperature
soon after one another; the relevant value was in units of percent.

## 3    Results and discussion

Representative examples of the surfaces of the non-seeded samples, seeded samples, and ice spheres
of three CsCl concentrations (0.5 M, 0.05 M, and 0.005 M) are displayed in Figure 2. These images were
recorded at the very beginning of the microscopic observations, and their surfaces are thus very little
affected by the sublimation and specific conditions inside the specimen chamber of the ESEM (e.g.,
low pressure and possible interactions with the electron beam); we therefore assume that the
observed features had resulted from the various freezing conditions. When BSEs are detected, merely
the material contrast is revealed in most cases. The white areas in the images represent the CsCl-
containing brine or CsCl crystals, while the dark or gray regions indicate the ice. Certain limited
topographical information is also obtained; it is visualized via various shades of gray. We measured
>50 samples and analyzed the acquired micrographs; from these, we then evaluated the impact of the
freezing method and the salt concentration on the appearance of the surfaces of the frozen samples,
location of the brine, and sizes of the ice crystals and grain boundary grooves.

**Method of freezing**

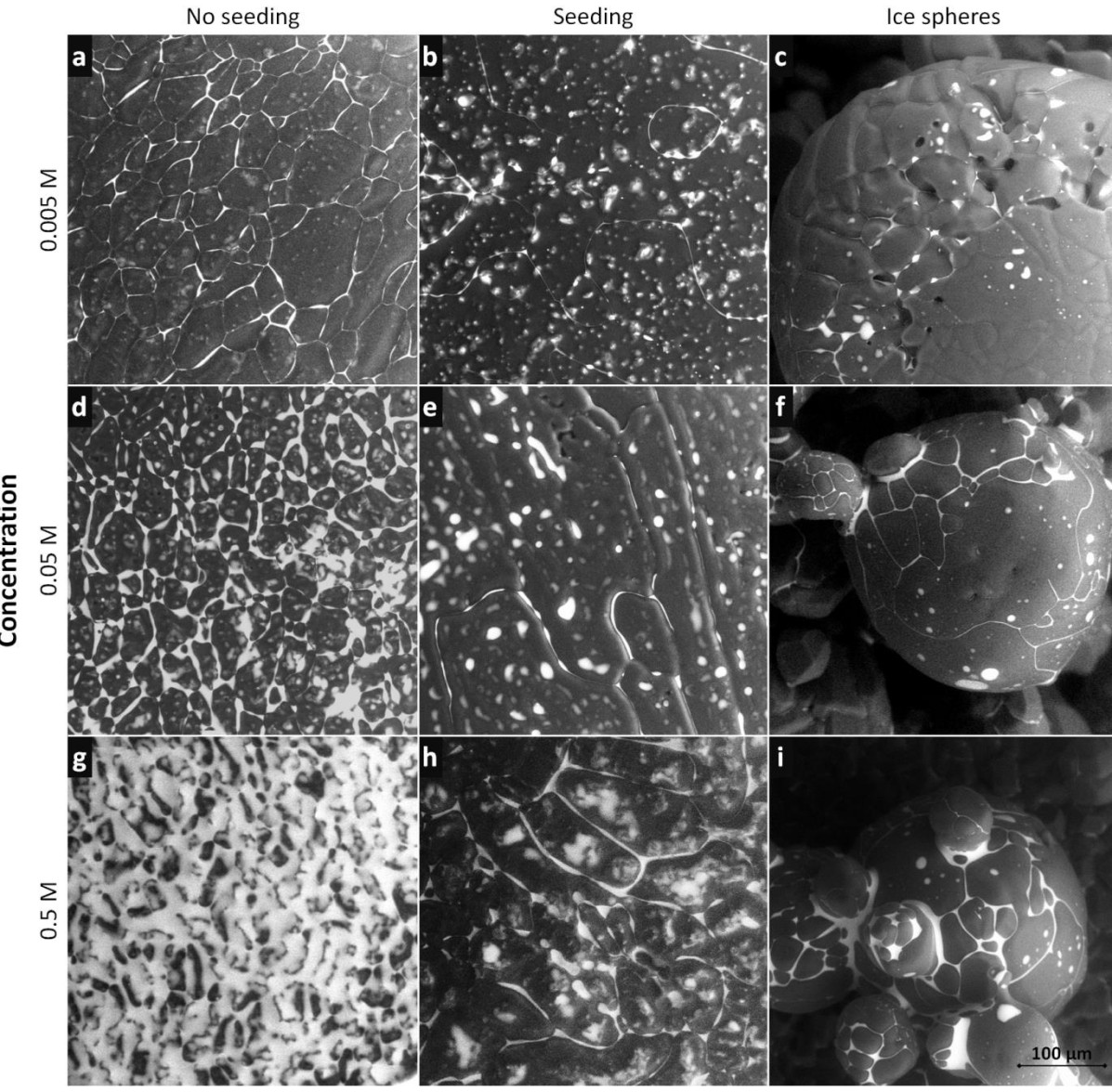

**Figure 2.** The ESEM images of the frozen samples prepared from the 0.005, 0.05, and 0.5 M CsCl solutions via spontaneously freezing a supercooled (non-seeded) droplet (a, d, g); freezing a droplet seeded with ice crystals (b, e, h); and spraying the solution into LN (c, f, i). The images were acquired at the very beginning of the observations, at the following temperatures: −22 °C (f, h, i); −23.5 °C (a-c); and −25 °C (d, e, g). The gray areas represent the ice, and the white regions denote the CsCl brine. Images of two additional samples for each freezing method and concentration are provided in the SI (Figures S7-S15).

### 3.1 Appearance of the surface of the frozen samples

**3.1.1 Non-seeded samples** (Figure 2, left column). The brine was located mainly at the grain boundary grooves. For the CsCl concentration of 0.005 M, the grooves were well visible; no brine puddles were located on the sample surface. As the concentration was increased to 0.05 M, the grain boundary grooves became thicker, and small brine puddles appeared on the surface. The surfaces of the samples (0.005 M and 0.05 M) were not smooth, with visible small humps. The surface of the most concentrated sample (0.5 M) was almost completely covered with a brine layer already at the beginning of the observations; during the procedure, the surface became completely brine-flooded due to the slow, gradual ice sublimation. Thus, the ice grains and grain boundary grooves could not be sufficiently recognized in this case.

**3.1.2 Seeded samples** (Figure 2, middle column). The amount of CsCl at the grain boundaries was apparently much smaller compared to that in the non-seeded samples: the grain boundaries were barely visible in the microscopic images of the 0.005 and 0.05 M seeded samples, and the surface of the 0.5 M seeded sample was not brine-flooded. However, brine puddles were abundant on the surfaces of the seeded samples. The surfaces were not smooth, with numerous humps formed (vide infra).

**3.1.3 Ice spheres** (Figure 2, right column). Spraying the solution into the LN led to the formation of ice spheres, with typical diameters ranging between 100 and 300 μm. Their prevalently regular and spherical shapes indicate that the droplets froze during levitation above the LN surface due to the inverse Leidenfrost effect: If they had frozen inside the LN volume, the shapes would have been pear-like (Adda-Bedia et al., 2016). The surfaces of the spheres were smooth, lacking visible humps and almost without brine puddles. For the concentration of 0.005 M, the grain boundaries were nearly invisible, exhibiting very little brine on the sphere surfaces. As regards the CsCl concentrations of 0.05 M and 0.5 M, the brine was freeze-concentrated in the grain boundary grooves and at joints between neighboring spheres.

## 3.2 Sizes of the ice crystals

The average size of the ice crystals on the surfaces of the frozen samples depended on both the freezing method and the solution concentration (Table 2). The largest ice crystals were detected in the seeded droplet (the average sizes of 105, 91, and 48 μm for 0.005, 0.05, and 0.5 M CsCl, respectively). These crystals exhibited oblong shapes, and therefore their longer and shorter dimensions were evaluated separately; the average dimensions determined are listed in Table 2. As the larger crystals often exceeded the field of view of the ESEM (Figure S2, distance F), the calculated sizes are biased towards lower values; thus, the average sizes should be somewhat larger. The smallest ice crystals were formed in the supercooled non-seeded droplet; the average sizes equaled 44, 33, and 26 μm for 0.005, 0.5, and 0.5 M CsCl, respectively. Medium ice crystals (the average sizes of 50, 55, and 39 μm for 0.005, 0.5, and 0.5 M CsCl, respectively) were observed on the surfaces of the ice spheres.

| Concentration of CsCl / M | Average diameters of the ice crystals / μm | | | | |
|---|---|---|---|---|---|
| | Non-seeded droplets | Seeded droplets | | | Ice spheres |
| | | overall | longer | shorter | |
| 0.005 | 44 ± 3 | 105 ± 11 | 135 ± 19 | 75 ± 8 | 50 ± 7 |
| 0.05 | 33 ± 2 | 91 ± 19 | 125 ± 25 | 53 ±8 | 55 ± 2 |
| 0.5 | 26 ± 2 | 48 ± 4 | 65 ± 6 | 31 ± 2 | 39 ± 4 |

**Table 2.** The average diameters (and their standard errors of the mean) of the ice crystals on the surface of the frozen samples prepared from the 0.005 M, 0.05 M and 0.5 M CsCl solutions via the three freezing methods. The ice crystals in the seeded samples had oblong shapes; therefore, the average sizes of the longer and shorter sides of the crystals were evaluated separately. The number of crystals used for the analysis is presented in the SI (Table S1).

In our experiments, the freezing of the seeded samples promoted the slowest freezing rates, as the ice crystals developed from a solution with a low degree of supercooling (the temperature of the droplet was −2 °C) and the progress of the crystallization front proved to be about 0.2 mm/s. This freezing technique produced the largest ice crystals. The freezing rates for the non-seeded samples (150 mm/s) and ice spheres were much higher compared to those in the previous case, and thus substantially smaller ice crystals had been produced. The ice crystals in the non-seeded samples and in the ice spheres were about half the size of those in the seeded samples.

Our examination of the large crystals at low supercooling is parallel to the findings by Macklin and Ryan, who investigated large crystal discs at low supercooling of pure water (Macklin and Ryan, 1966). The patterns in our micrographs show sizable disc-like crystals formed at low supercooling (Figure 2b), whereas 16-degrees Centigrade supercooling (Figure 2a) delivers numerous small crystals. The fact that faster freezing rates promote a high number of small ice crystals and slower ones form a few large ice crystals is well recognized and efficiently utilized by productive freezing and lyophilization techniques (Kasper and Friess, 2011;Jameel, 2010;Cao et al., 2003).

As mentioned above, the ice crystal size difference between the samples frozen under high and low supercooling was expected. However, we were rather surprised that the ice crystals in the non-seeded supercooled droplet appeared smaller than those in the ice spheres prepared by spraying a solution into the LN. Although LN boils at the very low temperature of −196 °C, its ability to quickly cool the sample down is reduced due to the inverse Leidenfrost effect (Adda-Bedia et al., 2016). An alternative reason for comparatively large ice crystals forming during the nebulization of a solution into LN may consist in a different freezing mechanism: at freezing rates from 210 to $2 \times 10^{-6}$ mm/s, morphological instability of the ice surface is predicted, whereas outside this range linear progression of the ice front is expected (Wettlaufer, 1992;Maus, 2019). The prediction was made for a 3.5% solution of NaCl, and varied in other compositions. The linear relationship between the freezing rate and the dimension of the formed lamellae holds only with temperatures of morphologically unstable growth; thus, if a high freezing rate induces linear growth of ice in ice spheres' preparation, the resulting ice crystal sizes cannot be used for comparison with the slower methods.

Another aspect affecting the crystal size consists in the concentration of the solutes. Generally, the ice crystals were smaller when the solute concentration had increased. The observation is in accordance with the conclusion proposed by multiple natural ice-core studies, namely, that high-impurity ice exhibits, as a rule, smaller grains than low-impurity ice at the same depth (Eichler et al., 2017). The explanation of this phenomenon may relate to the solution viscosity because ice crystal growth depends on the diffusion of the water molecules to the crystal surface, and an increase in the viscosity hinders the crystal growth (Braslavsky, 2015). The CsCl solution viscosity increases at concentration values higher than approximately 2 mol kg$^{-1}$ (Goldsack and Franchetto, 1977;Nakai et al., 1995). The concentration in the veins of the freezing CsCl solution is expected to be within such a range, as the eutectic composition corresponds to about 7.7 mol kg$^{-1}$ (Gao et al., 2017). A further factor influencing the ice crystal sizes possibly relates to a decreased heat capacity of the solution due to the presence of a solute (Bonner and Cerutti, 1976;Gao et al., 2017); thus, the solution may be cooled more effectively and the ice may propagate at a quicker pace, leading to smaller ice crystals. Changes in the above-described aspects could also cause altered microscopic convection patterns within the sample being frozen; these were previously shown to play a major role in the freezing of macroscopic quantities of sea water (Wettlaufer et al., 1997).

## 3.3 Brine on the surface of the frozen samples

**3.3.1 Effects of the salt concentration.** Typically, the brine on the surface of the frozen samples increased in amount with rising concentration of the salt. Such behavior is not surprising, as the maximum concentration of a salt in the brine and the proportion of the brine to the ice at each temperature are given by the liquidus curve in the phase diagram; the amount of the liquid brine increases with the salt concentration and temperature.

The grain boundary grooves became substantially thicker when the concentration had increased. The average widths of the brine-filled grain boundary grooves related to the concentration are given in Table 3. The measurements are displayed in S17. The average groove widths for the 0.005 M samples might be overestimated, as many grooves were so thin that they were almost invisible or their widths approached the image resolution (approx. 0.5 µm). The grain boundary grooves could not be

determined for the non-seeded sample with the CsCl concentration of 0.5 M, as the ice crystals were barely visible below the brine layer; the layer covered nearly the whole surface of the sample.

| Concentration of CsCl / M | Average widths of the grain boundary grooves / µm | | |
|---|---|---|---|
| | Non-seeded samples | Seeded samples | Ice spheres |
| 0.005 | 1.3 ± 0.5 | 1.1 ± 0.3 | 1.2 ± 0.3 |
| 0.05 | 2.1 ± 0.9 | 1.1 ± 0.3 | 1.9 ± 0.5 |
| 0.5 | ND | 5.7 ± 2.0 | 3.3 ± 1.1 |

**Table 3.** The average widths of the CsCl brine-filled grain boundary grooves for the frozen samples prepared from the 0.005 M, 0.05 M, and 0.5 M CsCl solutions via the three freezing methods. The uncertainties represent the pooled standard deviations. The numbers of independent samples used for the calculation are listed in the SI (Table S2). The average width of the grain boundary grooves for the 0.5 M non-seeded sample was not determined, because the brine layer had covered almost the whole surface of this sample.

**3.3.2 Impact of the freezing method on the surface coverage.** The brine surface coverage in relation to the freezing method was estimated for the frozen 0.05 M solutions. While the brine covered (18.2 ± 4.0) % and (9.0 ± 2.8) % of the surface in the non-seeded and the seeded samples, respectively, it occupied only (4.1 ± 1.5) % in the ice spheres (Table S3). These estimates are based on the detected relative brine surface coverage in 5 independent samples for each freezing method. However, the estimated surface coverage might be strongly dependent on the intensity of the detected signal, as discussed in the Methods section. Regrettably, neither the thickness of the brine layer nor the absolute amount of the salt on the surface can be directly evaluated using the ESEM; we can nevertheless estimate the relative amount of the salt on the surface of the sample, presuming that the thickness of the surface brine is similar to channels' width. Therefore, we estimated the relative amount of the salt on the surface of the frozen samples prepared from 0.05 M CsCl via the freezing methods I-III, exploiting the assumption that the thickness of the surface layer equals 1.5 µm. As a result, the relative amounts of the salt on the surface were 9, 4.5, and 26% for the non-seeded droplets, seeded droplets, and ice spheres, respectively. The calculations are provided in the SI.

The location of the CsCl brine within the sample and the related surface coverage were presumably dependent on the freezing rate and its directionality. In the globally supercooled non-seeded sample (cooled with a Peltier stage from the bottom), we suppose the crystallization was directed from the bottom upwards, as shown previously by Suzuki et al. (2007) for a supercooled water droplet on a silicon surface; thus, the salt was prevalently expelled from the bulk to the surface of the sample as the abrupt crystallization proceeded. Conversely, the seeded sample probably started to crystallize from the surface, close to the edge of the sample as the first place in touch with the seeding crystals. Even though small vertical temperature gradients were to be expected through the solution, we considered crystallization from the surface more likely, in accordance with a related study which had suggested that crystallization on the surface was $1 \times 10^{10}$ more probable than the corresponding process in the bulk (Shaw et al., 2005). Because the given freezing method promoted slower freezing rates, we speculate that most of the salt was expelled from the growing ice towards the bottom of the frozen sample. Thus, only a small portion of the salt could be seen on the surface. The lowest brine surface coverage was detected in the ice spheres created by the freezing of micrometric droplets from the surface inwards; however, due to the large surfaces of the ice spheres, the estimated relative amount of the salt on the surface was the highest among the techniques discussed. Nevertheless, a large portion of CsCl solution was expelled by the growing ice from the surface of the sphere towards the center, and the CsCl brine inclusions were trapped below the surface. Other arguments supporting this assumption are presented in the chapter discussing the interior of the frozen samples.

**3.3.3 Effects of the temperature.** As regards the seeded droplet of the 0.005 M CsCl solution, the brine surface coverage was studied as a function of temperature (Figures 3 and 4, Movie S1). The temperature of the sample reacts relatively slowly when the cooling stage temperature has been changed; therefore, the surface coverage variations are continuous, not stepwise. For this sample, about 20% of the surface was covered with the CsCl brine, initially at −23.4 °C. Note that such surface coverage is, unexpectedly, larger than that found on average even in the more concentrated seeded sample ($c$ = 0.05 M) within the previous section. The explanation may lie in the stochastic behavior of the freezing process; however, we identify the most important factor in the difference of the instrumental settings, which, for reasons already outlined above, does not allow quantitative comparison of the micrographs. The brine was located mainly on the ice surface humps (Figure 3); apparently, these had functioned as points around which the brine gathered and crystallized. To a major extent, the humps seem to be artifacts of the low pressure conditions in the microscope, as will be discussed in the following section.

When the temperature was gradually increased, the area of the brine-covered surface grew larger (Figure 4). Such behavior was expected, as the liquid fraction in the partially frozen sample increases with the temperature in the region between the $T_{eu}$ and the melting point. At the highest temperature examined, −12 °C, about 80% of the surface was covered with the brine. Subsequently, the temperature was abruptly lowered from −12 °C to −23.3 °C, and the brine surface coverage fraction decreased from about 80 % to approximately 40 %.

According to the CsCl-water phase diagram (Figure 1), the equilibrium concentration of the CsCl brine equals 7.7 mol kg$^{-1}$ at −23 °C and 3.9 mol kg$^{-1}$ at −12 °C (Gao et al., 2017), meaning that if a constant amount of the CsCl salt on the surface is assumed, the brine volume on the frozen surface should approximately double after the temperature had increased from −23 to −12 °C. Even though it is not possible to evaluate the volume from the microscopic images, we documented well that the brine surface coverage had risen four times during the warming, becoming much larger than the increase implied by the phase diagram. Presumably, the brine volume can be inferred indirectly from the surface coverage if spreading of the brine on the ice surface is governed mainly by the surface tension. We interpret the change in the brine volume as the central cause of the observed surface coverage variations, and we also suppose that the observed 4-fold rise in the coverage indicates an even larger volume alteration because the brine thickness will increase too. The strong relationship between the brine volume and surface coverage is indirectly demonstrated in Figure 4; the coverage follows the same trend as the relative amount of brine in the frozen sample ($w_{brine}$) calculated from the phase diagram using the formula

$$w_{brine} = \frac{m_{aq}}{m_{fr} - m_{aq}},$$

where, $m_{aq}$ is the molality of the aqueous solution of salt (that can be approximated by the concentration of the salt in case of low concentrations) and $m_{fr}$ is the molality of the brine in the frozen sample (it is dependent on temperature and can be inferred from the liquidus curve of the phase diagram). Thus, we are of the opinion that the change of the brine volume is the main cause of the surface coverage variation.

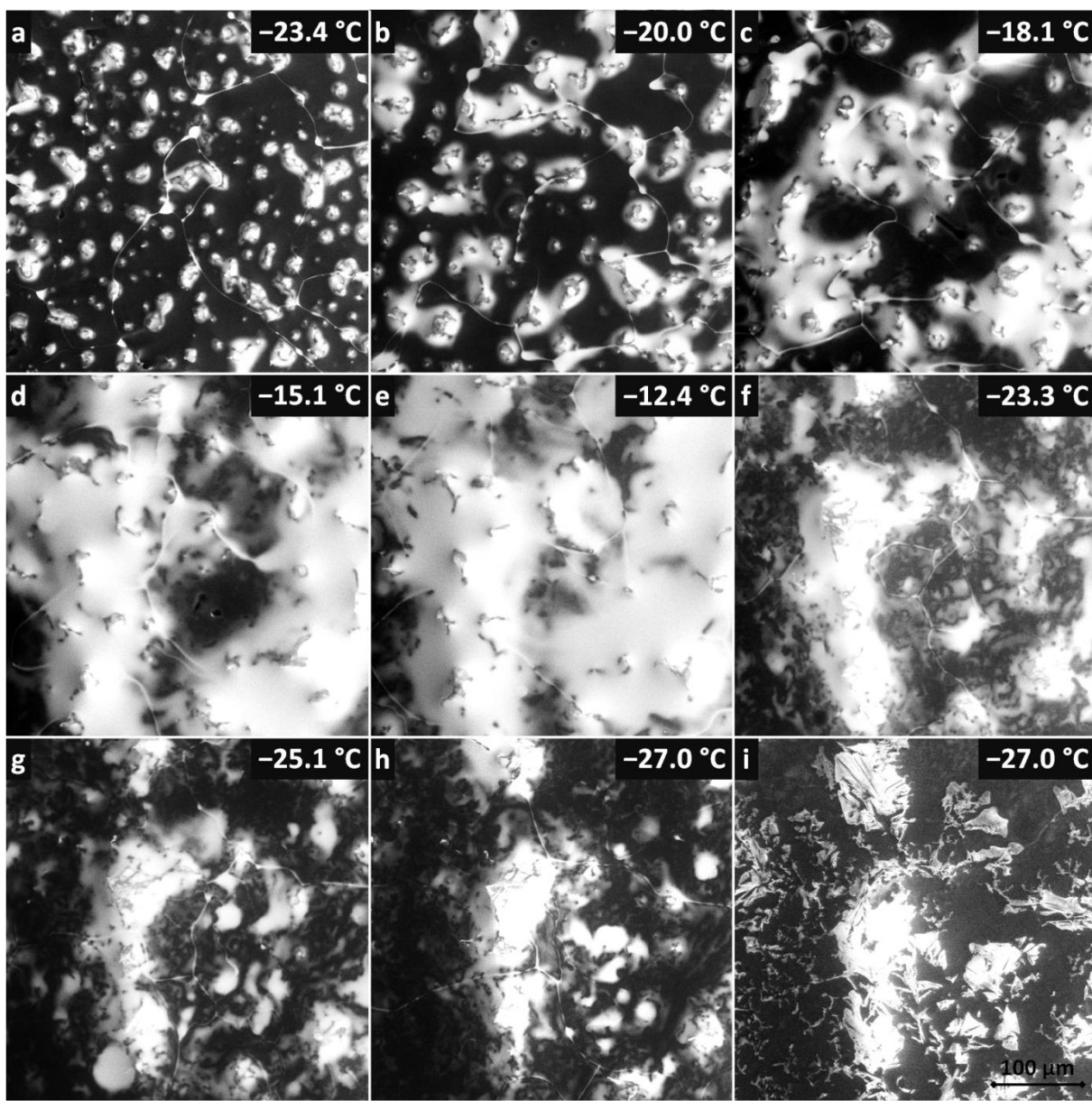

**Figure 3.** The brine surface coverage as a function of temperature for the seeded 0.005 M sample. The temperature was first increased stepwise from −23.4 °C to −12.4 °C and then decreased to −27 °C. The procedure took 30 minutes (Figure 4). A movie from this experiment is included in the Supplementary material section (Movie S1). The panels h and i display the salt crystals forming from the brine. The crystals formed along the grain boundaries and on the humps where the brine had accumulated previously. The crystallization started a minute after the sample holder temperature had been set to −27 °C. The air pressure in the microscope chamber was 500 Pa during the sequence. The scale in panel i is valid for all the images.

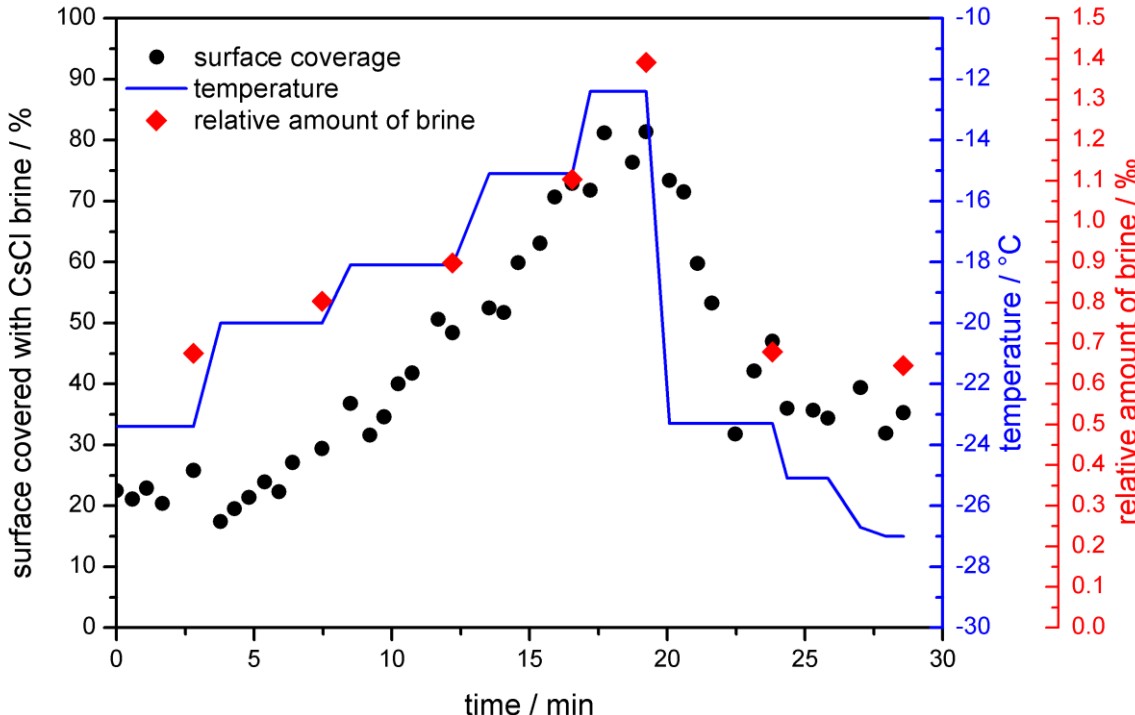

**Figure 4.** The relative brine surface coverage in the frozen seeded droplet (c(CsCl) = 0.005 M) at varying temperatures. The representative images used for the calculation are displayed in Figure 3. The temperature of the cooling stage during the measurement is indicated with the blue pattern. The red diamonds denote the relative amount of brine in the frozen sample, based on the phase diagram (Figure 1); temperatures 2 °C higher than those of the cooling stage were used for the calculation to better represent the sample temperature.

During the cooling phase, the decrease of the brine volume (and correspondingly also the surface coverage) by half was expectable, considering the phase diagram; however, the surface coverage by the end of the experiment was significantly larger compared to the initial stage, despite the essentially equal temperatures (−23.4 °C vs. −23.3 °C). Moreover, the temperature decrease down to −27 °C at the end of the experiment led to a surface coverage of about 30%, which, despite the temperature being 3.6 °C lower, exceeded the amount of coverage found at the initial stage by 10 %.

The results of this temperature cycling experiment indicate that, as the ice from the surface was required to melt during the heating in order to double the volume of the brine, a formerly inaccessible portion of the brine (restrained due to being trapped in the ice interior) surfaced. This increased the amount of the CsCl salt on the surface, and the brine volume rose dramatically above the expectations stemming from the phase diagram. The subsequent cooling of the sample lowered the volume by half, as expected, indicating that the overall amount of the CsCl salt on the surface had not changed at the cooling stage. This observation implies that the temperature cycling may result in enhanced accumulation of the brine on the frozen surface during the heating phase.

In the experiment, the CsCl started to crystallize from the brine at −27 °C only after a time delay (Figure 3i). The salt crystallization front propagated along the brine-filled grain boundaries and puddles (Figures 3, S3, Movie S1). The salt crystals exhibited fine dendritic structures and remained at the humps for a longer period. The brine left on the surface produced voids in the ice bulk; similarly, vertical, gravity-driven brine drainage is known to cause sea ice density to decrease (Timco and Frederking, 1996). We are currently investigating the details of salt crystallization and ice sublimation in salt-containing samples under various conditions.

In other experiments, the cooling stage temperature of −25 °C was sufficient for the brine to
crystallize after a prolonged period. The hitherto published $T_{eu}$ values of the CsCl-$H_2O$ mixture range
from −24.83 to −22.3 °C (Cohen-Adad, 1991;Dubois et al., 1993;Fujiwara and Nishimoto, 1998;Monnin
and Dubois, 1999;Chen et al., 2005;Gao et al., 2017). The thermal camera measurements in
atmospheric conditions (without the reduced pressure environment of the ESEM sample chamber)
showed that the sample was about 2 °C warmer than the cooling stage. Further warming of the sample
during the ESEM experiments was to be expected, especially due to the effects of the electron beam
and the relatively warm gas purged through the specimen chamber; conversely, the surface cooling
embodied an expectable process too, considering the ice sublimation and water evaporation from the
brine. These factors could contribute to the salt crystallization on the sample surface, even when the
temperature of the holder does not change; still, the overall temperature difference between the
surface of the frozen samples and the cooling stage will likely not exceed 2 °C.
The observed ice surface coverage changes have consequences for the interaction with radiation
under natural conditions: the higher the brine volume on the ice surface, the more absorbing the
surface compared to the condition of dry ice surface, which is highly scattering (Light et al., 2009;Light
et al., 2003). Thus, the temperature increase with the subsequent spread of the brine on the surface
may lead to the surface darkening, resulting in higher solar radiation absorption and further increase
of the temperature. Conversely, the brine crystallizing on the ice surface would result in larger
reflection of the radiation, as the crystals would scatter the light substantially more effectively
compared to the liquid brine (Carns et al., 2016;Light et al., 2016). These effects, together with the
number and sizes of inclusions, ice grains, and location and characteristics of the absorbing particles,
influence the overall absorption properties of icy bodies (Warren, 2019).

### 3.4 Formation of the humps
The surface humps were observed in the numerous non-seeded and seeded samples, often in co-
location with the brine puddles. The time-lapse images revealed the humps forming at places where
the puddles had been located previously (Figure 5). The humps usually exhibited broader bases and
thin tips; the brine was located mainly at the bases, not on the tips (Figures S4, S5). As the sublimation
continued, the tips became even thinner. In some cases, the thin tip separated from the body of a
hump and fell off (Figure S4). From these observations we infer that the humps are predominantly
formed during microscopic examinations due to ice sublimating from the frozen samples; we do not
expect the humps to be originally present on the surface of the samples. The formation of the humps
appears to be facilitated by the brine puddles on the ice surface: When the surface is covered with the
brine, the sublimation from the given spot is retarded, and a hump is formed. Such formation might
be an artifact caused by experimentation in a low-pressure environment. Previously, similar
observations had been explained in terms of charged peaks formed by etching in the SEM (Barnes et
al., 2002).

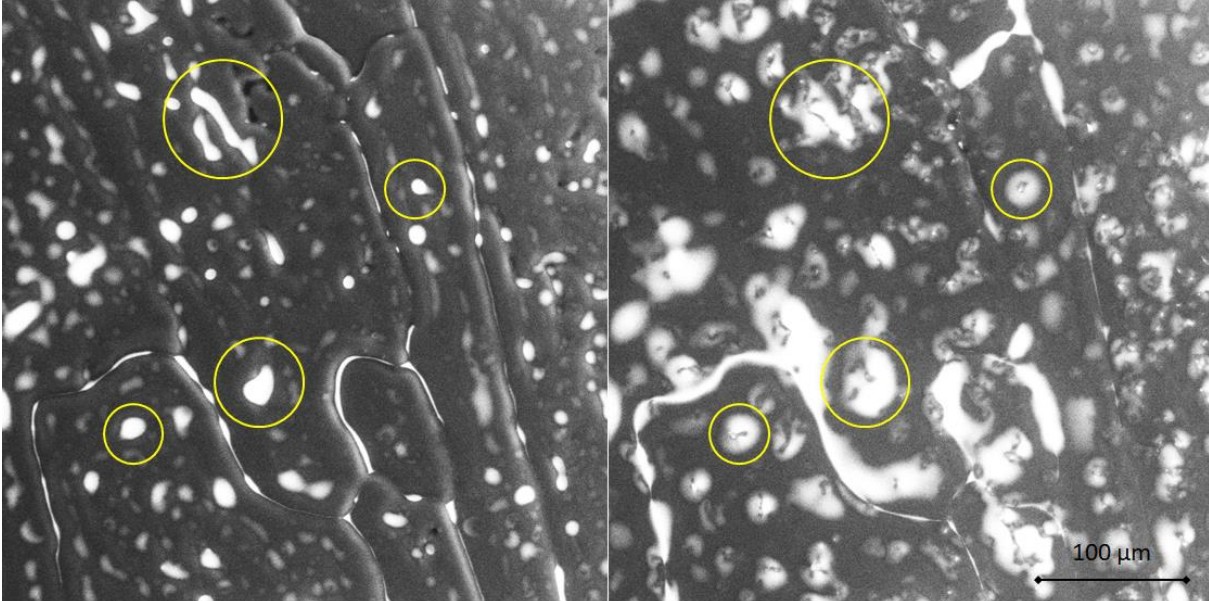

**Figure 5.** The ESEM images of the seeded sample prepared from the 0.05 M CsCl solution at the very beginning of the observation (left panel) and after 90 s (right panel). The images were recorded at the temperature of −25 °C. As conditions below the sublimation curve had been set inside the specimen chamber, sublimation of the top layer of the ice was presumed during the time interval between two images. It is clearly visible that the humps formed at spots previously covered with the brine; four examples of such spots are encircled.

## 3.5 Dynamics of the ice surface

The dynamics of the frozen samples' surfaces could be observed even at −23 °C. The CsCl brine-filled grain boundaries were not static: their positions changed swiftly in time. Interestingly, the positions of the humps on the ice surface did not change accordingly. A representative example is shown in Figure 6 and Movie S1. The surface of the seeded droplet from the 0.005 M CsCl solution was observed at −23.4 °C. As is demonstrated by the yellow arrow, whose position remains constant in all the panels in Figure 6, the indicated hump was originally encircled by brine-filled channels (panel a). We suppose the channels denote the ice grain boundary positions. With the time progressing, the grain boundary on the right-hand side performed an apparent gradual movement through the hump, while the hump did not change. The brine channel moved from the spot marked with the arrow tip to that marked with the red dot in Figure 6d (distance 31 μm, time 1:40 min), thus advancing by approximately 19 μm/min. It was also observed that the humps had caused the directional propagation of the ice boundaries to decelerate; several such examples are presented in Movie S1. A larger brine volume surfaced at the triple junctions of the ice grains. The dynamics of the process, together with the sampling relatively slow compared to the ice boundary movement, lend the brine reservoirs the appearance of propagating comet-like objects; this effect is demonstrated via the related encircled points in Movie S1. These brine pools occasionally appear at the locations of caverns formed by bubbles, thus delineating the grain boundary, which would otherwise be difficult to observe.

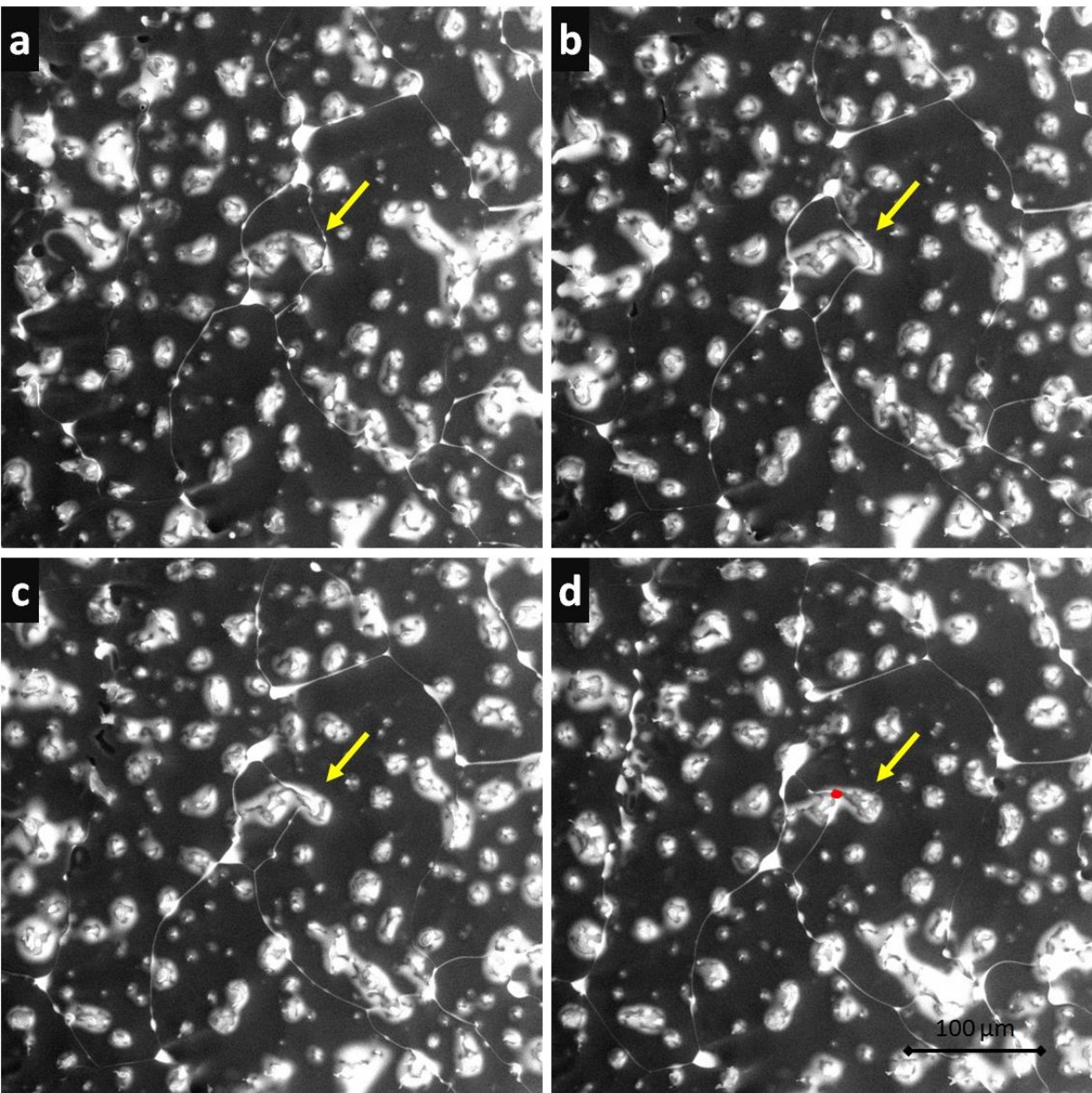

**Figure 6.** The ice surface dynamics observed at −23.5 °C and 500 Pa. The brine-filled grain boundaries (white channels) shifted their positions in time, while the positions of the humps did not change. An example is indicated by the yellow arrow pointing to a CsCl-filled channel moving through a hump. The arrow indicates the initial position of the brine channel in all of the figures. The sample was prepared from the 0.005 M CsCl solution by freezing a droplet seeded with ice at −2 °C. The distance between the brine channel (red point) and the arrow tip (where the movement started) is 31 μm; the channel moved by ~19 μm/min. The time interval between the images is approximately 30 s. The scale in panel d is valid for all of the images.

The fact that ice is a very dynamic medium allowing reconstitution at sub-zero temperatures was previously evidenced using, for example, time-lapse X-ray tomography; up to 60 % of the total ice mass recrystallized during 12 hours at − 15 °C (Pinzer and Schneebeli, 2009;Hullar and Anastasio, 2016). Our research team is not the first one to have observed disjoint behavior of the ice surface in relation to the underlying ice crystals; fresh ice growing in steps of 100 − 4,000 nm to traverse the ice grains regardless of the boundaries was already noticed for planar ice growth (Ketcham and Hobbs, 1968). Moreover, the observed mutual interactions between the surface humps and the ice grain boundaries causing temporal retardation of the boundaries and partial draft of the hump (in the direction of the boundary motion) can be classified as Zener slow-mode pinning (Eichler et al., 2017). New ice grain formation (probably ascribable to the grain-boundary bulges mechanism) and the subsequent ice

crystal growth are visualized in Figure 7 (Steinbach et al., 2017). In this context, however, another explanation is also acceptable, as discussed below.

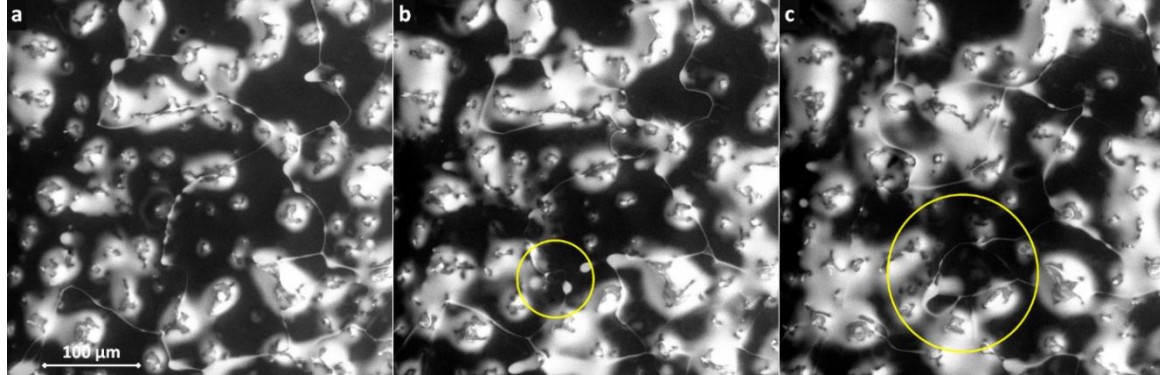

**Figure 7.** The ice surface dynamics observed at −23.5 °C and 500 Pa; newly formed ice grains are encircled.

The actual reason why the brine channels were dynamic while the humps remained static is not entirely clear to us. The causes of the recrystallization process are nevertheless appropriately summarized by Steinbach; the relevant factors include the thermal gradients and strains in the samples (Steinbach et al., 2017). Two possibilities are considered in this respect:

Firstly, the movement of the brine channels could be induced by a temperature gradient across the surface of the frozen ice. The sample was prepared from a hemispherical droplet by cooling from the bottom. Thus, the height was most likely not constant across the sample and would be the most prominent in the center, allowing the temperature gradient to rise across the surface. A thermal camera was used to estimate the temperature gradients throughout the sample during cooling outside the ESEM chamber; it was not experimentally feasible to insert the camera into the chamber. As the temperature of the sample was dropping from 0 to −18 °C, the temperature at the center of the surface was approximately 2.7 °C higher than at the periphery. This difference had been caused presumably by the greater thickness of the sample at the central section. However, after the sample was cooled down and its temperature became approximately constant (which typically took about 3 minutes), the gradient was reduced, albeit in an opposite manner: The center was about 0.2-0.3 °C colder than the periphery, producing a small temperature gradient throughout the sample even after the cooling. Regrettably, as the experiment was not conducted in the ESEM chamber, the effects of the gas flow on one hand and sublimation on the other are not included. Under a temperature gradient, the solute would exhibit higher concentration at the colder end of the brine inclusion and lower concentration at the warmer end, resulting in migration of the brine inclusions toward the warmer region of the ice (Light et al., 2009). The behavior observed in the present study indicates that the water molecules at the brine-filled grain boundaries are redistributed much faster than those on the ice surface. The dissolved salt may diffuse through the veins to increase the salt concentration on one side and decrease it on the other. Where the concentration increases, the ice grain boundaries will melt, and where it decreases, the ice will grow to maintain equilibrium conditions.

The sample presented in Figure 6 was sufficiently cooled down, as enough time had elapsed since the last temperature adjustment; therefore, we assume there might be a temperature gradient of only several tenths of °C across the sample.

A second hypothesis to explain the apparent movement of the brine channels focuses on the ice sublimation from the sample surface. The brine channels were not necessarily perpendicular to the surface; the angle between the surface and the channels might have been low, and therefore the sublimation of a thin layer of ice from the surface could theoretically result in apparent movement of the brine channels while the appearance of the humps would not change markedly. Similar reasoning had been employed for the sublimation of not perpendicularly oriented ice crystals to explain the observed excessive width of the ice-grain grooves (Cullen and Baker, 2001). The sublimation rate largely depends on both the amount of water vapor purged into the chamber and the temperature.

We do not know the complete time required for our samples to sublimate, because the process was
not aimed at in our experiments; contrariwise, we intended to prevent sublimation of the samples. We
usually imaged the samples for about 30 minutes, and they never sublimated during this period. The
sample height at the start of the imaging corresponded to approximately 1 mm (based on the working
distance of the ESEM). The upper limit of the (vertical) sublimation rate can thus be calculated if full
sublimation of the sample is assumed to materialize within 30 minutes; the (vertical) sublimation rate
would then be 33 μm/min. The brine channel (Figure 6) moved by ~19 μm/min. If the apparent
movement of the brine channel is caused solely by the sublimation, the angle between the channel
and the horizontal line will be 60°.
We are, however, uncertain if the reason for these unusual dynamics of the frozen surface consists
in one of the previously described conditions, both of them, or a completely different process.

## 3.6 Interior of the frozen samples


**3.6.1 Sublimation.** The subsurface structure of the samples was revealed after partial sublimation of
the ice from the surface made from the 0.005 M CsCl solution (Figure 8). Subsurface hollows were
observed in all types of samples, namely, the non-seeded and seeded ones as well as the ice spheres.
The numbers and sizes of the hollows nevertheless differed: the hollows in the non-seeded samples
were small (units of micrometers) but numerous; those in the seeded samples were scarce; and those
in the LN-prepared ice spheres were numerous, with the size varying from units to one hundred
micrometres. The formation of these hollows is presumably associated with air bubbles trapped below
the ice surface. For all the freezing methods, opening the hollows by further sublimation often revealed
them to be partially filled with the brine (Figure 8); the co-location of the brine with gases suggests
that the brine in the veins is air-saturated. The presence of gases in the brine had been shown
previously to ease the ice recrystallization and boundary migration (Harrison, 1965). Recently, brine
inclusions trapped below the ice surface co-located with air bubbles were found (Hullar and Anastasio,
2016). The co-location of the brine and the bubbles was most pronounced for the ice spheres, formed
by freezing microdroplets of the solution by the LN at temperatures close to 77 K. The ice spheres were
most likely frozen from the surface inwards; therefore, the entrapment of the brine and air inside the
structure was anticipated, as was also observed  in fractured samples in a similar manner (McCarthy
et al., 2013).  Conversely, the seeding method allows slow ice crystallization, leading to major expulsion
of the air either out of the solution or downward, towards the bottom; we are nevertheless unsure of
the details of such scenarios.

**Method of freezing**

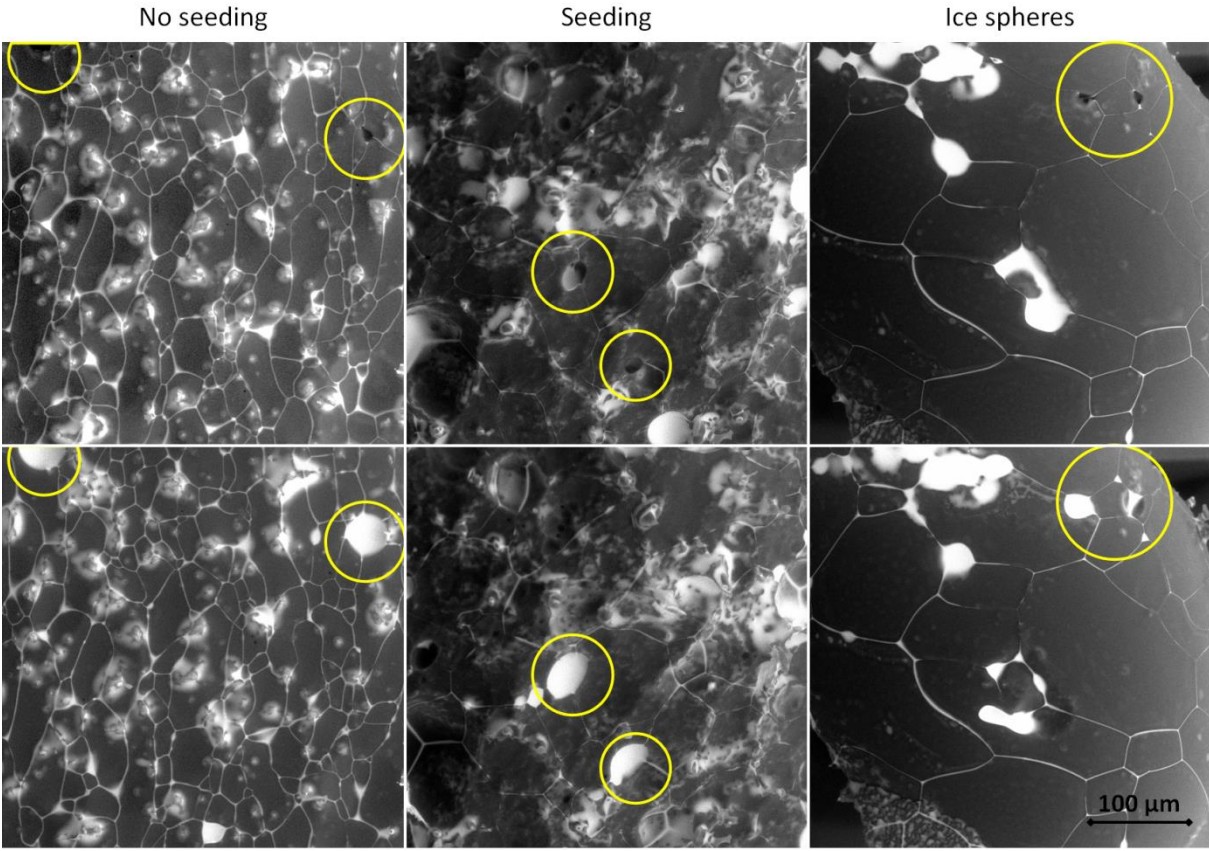

**Figure 8.** The ESEM images of the partially sublimed frozen samples prepared from the 0.005 M CsCl solution: the non-seeded droplet (left, −25 °C); the seeded droplet (middle, −25 °C); and the ice sphere (right, −23 °C). The hollows in the internal structure were revealed after the initial sublimation of the ice from the surface (upper panels). Further sublimation showed the hollows filled with the brine (lower panels); the time interval between the upper and the lower images was about one minute. The gray areas represent the ice, while the white regions indicate the CsCl brine.

**3.6.2 Fragmentation.** An additional technique to observe the interior of a frozen sample consisted in fragmenting the LN-frozen macroscopic ice samples and inspecting the pieces originated from the given interior in the ESEM. Representative images of the fragments are displayed in Figure 9; it is clearly visible that the ice and the brine are arranged in parallel lamellae in the original interior of the fragmentized sample. The brine lamellae appear as lines on the micrograms. These lamellae were approximately equidistant within one facet of the fragment; however, when various facets are compared within one sample, the distances between the lamellae might differ significantly. The closer brine lamellae spacing for the lower-right facet in Figure 9b was approximately 10 μm; the wide-spaced arrangement (at 40 μm) is displayed on the upper-left facet. The lamellae were long and narrow. Sparsely, the parallel structure of the lamellae was interrupted by the bridge connecting two adjacent brine layers across the ice plate; several examples of these are encircled in Figure 9a. Only few bridges were typically recognized on the individual facets. We cannot determine if these features were present in the samples naturally or through the sample fragmentation (Maeda et al., 2003).

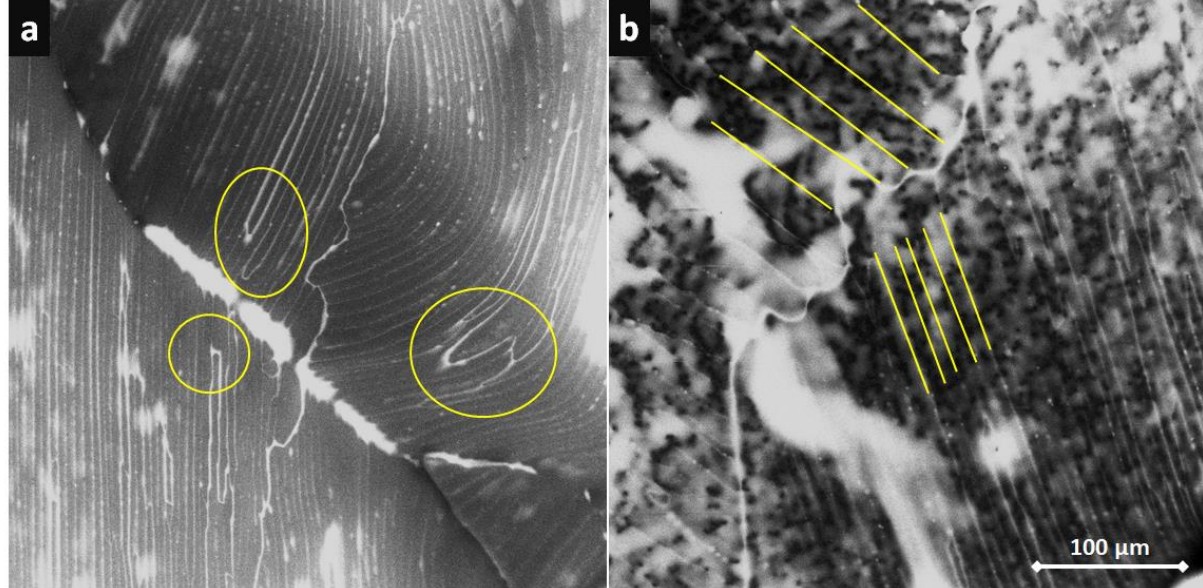

**Figure 9.** The interior of the fragmentized LN-frozen sample of 0.05 M CsCl imaged with the ESEM at −25 °C. The gray areas represent the ice, and the white regions indicate the CsCl brine. The brine was arranged in densely spaced parallel lamellae. Bridging of the ice was rare within a facet; examples of the effect are highlighted in panel a. The brine lamellae were equidistantly spaced within a facet. However, the distances between the lamellae might differ significantly among the facets (b). Images of two additional samples for this freezing method are provided in the SI (Figure S16).

The structures observed in the interior of the LN-frozen samples resembled the lamellar eutectic structure of alloys (Losert et al., 1998;Yan et al., 2017) or the pattern often recognized when sea ice is formed (Anderson and Weeks, 1958;Petrenko and Whitworth, 1999;Weeks, 2010;Thomas, 2017;Nagashima and Furukawa, 1997;Shokr and Sinha, 2015). This pattern is generated due to the constitutional supercooling of the liquid ahead of the freezing interface. It has long been known that the fastest freezing produces dendritic structures, whereas gradual slow freezing is characterized by cellular and planar structures, respectively (Rohatgi and Adams, 1967;Trivedi and Kurz, 1994). The linear solidification region  is reached also upon very fast freezing (Wettlaufer, 1992).  As a matter of fact, the microscopic structure of ice and brine is a complicated function of the thermal gradient and rate of freezing (Losert et al., 1998;Yan et al., 2017;Maus, 2007). The spacing of the parallel lamellae of pure ice separated by brine layers, created in a direction perpendicular to that of freezing, is characteristic of particular freezing rates and salt identities and concentrations. In terms of freezing salts solutions (NaCl, KCl and LiCl), the spacing between the perpendicular lamellae was experimentally found to be proportional to the inverse square root of the freezing rate (Rohatgi and Adams, 1967). The same paper reported that the maximum freezing rate, as well as its average, was inversely proportional to the square of the distance from the chill. At the same time, an increase of the solute concentration is responsible for the larger spacing between the lamellae. It is therefore conceivable that the uneven spacing observed in Figure 9b is due to unequal temperatures and/or concentrations for the growth of the individual ice facets. Another contributing factor could be the altered relative angle of the facets towards the detector, which would result in biased observed spacing. Using the theoretically predicted instability for the 0.035 % NaCl solution, (Wettlaufer, 1992) we can conclude, for the observed 10-μm spacing, that the freezing rate can range from 0.1 to $10^{-5}$ mm/s.  Thus, the freezing of the 16 °C supercooled sample can proceed 1,000 times (or more) faster than that of the LN-immersed capsules.

It is interesting to note that lamellar brine arrangement was not observed in any other sample, namely, neither on the surface of the LN-frozen macroscopic sample (Figure S6) nor on or below the surface of the ice spheres (Figures 2 and 8, right-hand columns). Instead, structures resembling a

cellular arrangement rather than a lamellar interface are formed in the small samples (Weeks,
2010;Losert et al., 1998). We can only speculate if the absence of lamellar growth in the micrometric
ice spheres is due to the higher freezing rate or the exposure to a lower temperature gradient
compared to the macroscopic sample with the diameter of 8 mm immersed in LN. Two significantly
different ice sample structures were observed, even though LN was used as the coolant in both cases.
A recent study of the inverse Leidenfrost effect of levitating drops on the surface of LN revealed strong
dependence of the freezing time on the size of the droplet (Adda-Bedia et al., 2016); therefore, the
size and surface properties of micrometric ice spheres in contrast to those of gelatine capsules can be
among important contributing factors as regards the described difference.

## 4   Relevance to previous observations
The ESEM observation presented herein employed a proprietarily developed detection system
allowing the use of very low currents (40 pA) and short dwell times (14 µs) to minimize the samples'
degradation during the imaging (Nedela et al., 2018). The electric field was not found to influence the
ice growth (Rohatgi et al., 1974). There was a subtle magnetic field present in our microscopic
chamber. Thus, our observations differed from the natural conditions mostly due to the low pressure
of 500 Pa inside the chamber.
In the experiments, CsCl was used as the solute thanks to its high contrast between the caesium
atom and the oxygen and hydrogen of ice when BSEs are detected. It served as the model compound
for monovalent chlorides, abundantly present in the environment. CsCl solutions exhibit properties
relatively similar to those of NaCl ones; in particular, the eutectic temperatures important for this study
are very close to each other ($T_{eu}$ (CsCl) = -23 °C, $T_{eu}$ (NaCl) = -21.21 °C) (Brady, 2009). The tested
concentrations ranged over two orders of magnitude. The values within 500 - 50 mM define the
concentration of NaCl in seawater, and therefore they are also descriptive of fresh sea ice in the given
context (Massom et al., 2001;Thomas, 2017). NaCl concentrations reaching up to 160 mM were
detected immediately next to a highway treated against road icing; 50 mM of a salt solution can thus
be considered a concentration potentially found farther from roads or also in their close vicinity when
the salt was already partly flushed away (Notz and Worster, 2009;Labadia and Buttle, 1996). The values
discovered in surface snows in Arctic coastal regions or on the surface of frost flowers span between
10-100 mM but may also be as low as 5 mM (Beine et al., 2012;Douglas et al., 2012;Maus, 2019). A
similar concentration was used in the report on solute locations in ice by Hullar and Anastasio (2016).
The principal finding presented within our study is embodied in the very strong sensitivity of the
ice-brine morphology and brine distribution to the freezing method (Figure 2, 3, 9). Even for identical
solution concentrations, the method and direction of freezing strongly modify the appearance of the
ice surface morphology. In all of the experiments, we observed a clear difference between the
interconnected system of the brine in the grain boundary grooves and the predominantly separated
brine puddles. At the lowest salt concentrations, free ice surfaces dominated. As the concentration
rose, the grain boundary grooves broadened, and both the amount and the size of the brine puddles
increased; however, some parts of the surface would remain bare even at the highest concentrations.
Partial surface wetting had been invoked previously (Domine et al., 2013).
The directionality of freezing influences to a large extent the amount of brine exposed on the
surface. Freezing from the outside inwards segregates the brine inside the ice matrix, whereas the
same process starting from the bottom expels the brine to the ice surface. The directionality of freezing
was previously shown to play a key role in determining the final distribution of impurities at the
macroscopic scale for the formation of ice on seawater (Wettlaufer et al., 1997). With the samples
cooled from the top by air at -20°C, it was observed that the salt is retained in the mushy layer of ice
until it reaches a critical height. After this stage, plumes (capsules) of a saturated salt solution (having
a higher density and low temperature) migrate downwards, leaving space for a less concentrated
solution and thus faster growth of the ice. Such a procedure then reduces the sea ice density (Timco
and Frederking, 1996). Our freezing methods differ from that related to natural sea water in the rates
and directionality. By extension, we would like to imply that the role of material convection in relation
to the drop size should be considered for the modeling of the freezing process.
The distinct ice grain sizes and textures shown for the various freezing methods (Figures 2, 8, 9,
and Table 2) indicate possible differences in the mechanical, thermal, and optical properties of the ice,
influencing, for example, the samples' permeability for percolating solutions or their optical properties.
Related changes could affect numerous properties of ices, having geological consequences such as the
impact on sea ice desalination and melting; further, the variations might influence also the energy
balance of ices and aerosols and rates of the (photo)chemical reaction (Thomas, 2017). In our
experiments, the samples were invariably placed on the silicon pad in the microscope. This procedure
renders the observation different from that of natural sea ice, where the brine from the ice may leak
into the seawater (Notz and Worster, 2009). Nevertheless, our observations can be considered a good
model for salty ices on solid grounds.
The ice sublimation effects displayed in Figure 8 suggest the character of the behavior of salty ice
samples exposed to dry winds in the nature, namely, samples in which gradual brine accumulation is
expected.

## 4.1 Connection to previous microscopic observations
In previous studies of frozen aqueous solutions, two regions were described: one of the
uncontaminated, pure ice matrix, and the other with impurities concentrated in liquid-like
compartments above their eutectic temperatures (Krausko et al., 2014;Thomas, 2017;Lake and Lewis,
1970;Rohatgi and Adams, 1967). The impurities' inclusion was observed via several microscopic
methods for both laboratory-prepared samples and natural ice and snow. However, the imaging
methods reaching -20°C or more did not have a resolution large enough to visualize the impurities in
the grain boundary grooves and were blind to the ice topography. For this reason, the impurities in the
grain boundary grooves of polycrystalline ices were then observed in high vacuum via low temperature
(< -80 °C) SEM (Barnes et al., 2003;Cullen and Baker, 2001;Blackford et al., 2007;Rosenthal et al., 2007).
Our technique facilitates the observation of all the aspects: the liquid brine puddles, brine in the grain
boundary grooves, and, to some extent, ice topology. The aim in this context is to summarize the
previous observations, with a focus on the dimensions of the observed impurities' elements; first,
attention is paid to the veins of subtle impurities and grain boundary grooves and, second, the larger
puddles are discussed.

**4.1.1 Observing the grain boundary grooves and veins**
Typically, in the procedures adopted to observe the impurities, the ice samples were repeatedly
exposed to "etching", e.g., ice sublimation at a higher temperature (-20 °C)(Rosenthal et al.,
2007;Cullen and Baker, 2001). The analysis allowed examination of pure ice crystals surrounded by
veins containing the impurities. Using the energy dispersive X-ray technique, the veins were found to
contain elements (e.g., Cl, S, Na, Mg) corresponding to the common minerals (NaCl, sulfates) for
natural glacier samples (Cullen and Baker, 2001). At certain instances, the grain boundary impurities
would coagulate to form filaments that can eventually peel off the ice due to differing thermal
expansion coefficients. The diameters of the filaments varied between 0.2 and 2.5 μm, and the
dimension of the nodes (the meeting points of four grains) was approximately 5 μm (Blackford et al.,
2007;Rosenthal et al., 2007). However, the diameter of the groove formed by the sample sublimation
at 253 K for 8 weeks reached as high as 250 μm. The apparent size of the sublimation groove can be
biased by the fact that the ice crystals do not need to be oriented perpendicularly to the plane of
observation. The observation most relevant to the ice spheres presented herein is that of the NaCl
solution (0.043 M) nebulized into LN and observed at -120 °C (Blackford et al., 2007). The newly
prepared ice spheres exhibited sub-micrometer fine filaments of salt on the surface, whereas sintering
at -25 °C allowed separated patches (up to 100 μm in diameter) and filaments threading the ice crystals
due to metamorphism (Blackford, 2007;Domine et al., 2013). Further sublimation of the ice striped the
filaments and showed that they form 3D interconnected structures, indicating also that each filament
has a star-like geometry, which is appropriately explained via considering the surface energies of the
ice and the brine.
All the dimensions of the veins observed previously correspond very well to those of the grain
boundary grooves reported in the present paper (Table 3). The grain boundary grooves are formed at
positions where two neighboring ice grains meet on the ice surface; thus, they thread each individual
ice grain. Apparently, the amount of the brine on the ice surface was not sufficient to fill the groove
around all the ice grains due to low brine concentration and/or the freezing method concentrating the
brine towards the interior of the ice matrix (Figure 2c, e, f). Even for the most concentrated samples,
the grain boundary grooves were found to range below 6 μm on average. The ice grain boundary
grooves were predicted to increase their width with rising temperature and upon aging; (Nye, 1991)
for example, under certain conditions, the values of 0.3 mm and 0.7 mm are assumed after 10 and 100
days, respectively, at 0 °C.
**4.1.2 Observation of the brine puddles**
The advantage of optical microscopic methods and X-ray tomography consists in the possibility of
working at atmospheric pressure and an arbitrary temperature; the central drawback of these
approaches rests in the compromised spatial resolution (Blackford, 2007). Importantly, optical
microscopy maintains the transparency of ice to visible light, causing the image to include the
subsurface information (Eichler et al., 2017); this aspect can be considered a benefit or a drawback,
depending on the situation. Thus, the applied method relied on imaging the larger brine patches, but
the brine in the subtle ice boundary grooves could not be represented.
The fluorescence microscopy technique exploiting a pH indicator showed a major difference
between frozen water and a 0.1 mM solution of NaCl at -7 °C (Cheng et al., 2010): the former left pools
of the solution around the ice crystals (approx. 12 μm wide), whereas the latter was specific in that the
solute was mostly rejected towards the edge of the microscopic field of view, leaving only a few thin
brine channels in the bulk of the ice. Another study based on fluorescence microscopy was intended
to assess the amount of aggregation of bovine serum albumin labelled by a fluorescence probe (Roessl
et al., 2015). Fast freezing rates produced more puddles of proteins, while slow ones led to fewer larger
patches (millimeter sized), with the protein contained in a more concentrated solution. Such results
agree with those assessing the extent of aggregation via the spectroscopy of methylene blue (Heger
et al., 2005). The patches containing various salts were shown, via X-ray fluorescence imaging, to
increase their surface concentrations with decreasing temperatures (Tokumasu et al., 2016). In the
referenced study by Tokumasu et al., the patches were determined to be about 30 μm wide at
temperatures from -5 to -15 °C. The optical microscopy and Raman spectroscopy of ice core samples
detected microscopic impurities assigned to salt crystals and dust scattered through the ice matrix
(Eichler et al., 2017). Eichler et al.'s report did not find any signal for liquid inclusions in the form of
pools or veins around the ice grains at -15 °C. The resolution of the microscopy corresponded to 3
μm/pixel, a value higher than the dimension of most of the grain boundary grooves measured in our
experiments. Therefore, it appears probable that the method did not facilitate discerning the ice grain
boundaries (Eichler et al., 2017). Recently, the Raman microscopy of ice revealed the dependence of
the brine puddles' width on the temperature; the width increased from 10 μm at -20 °C to 20 μm at -
10 °C for 0.6 M NaCl solutions (Malley et al., 2018). Interior images of laboratory-frozen CsCl aqueous
solutions (1 mM) were acquired with X-ray microtomography (Hullar and Anastasio, 2016). Slow
freezing in a refrigerator led to the formation of large pools of impurities (tens to hundreds of
micrometers), whereas dumping the sample into liquid nitrogen produced small segregations, namely,
ones sized below the voxel size of 16 μm. The impurities were often associated with the air bubbles.
The ice grain boundaries were not visualized as their dimensions are below the resolution of the
applied method. Quantification of the overall amount of the CsCl present in the samples reveals that
a substantial part of the CsCl is not detected in the analysis; thus, it must be present in the sub-
detection compartments in the bulk of the ice matrix. Based on the current study, we suggest that the
amount of missing brine can be located in the veins. A similar conclusion was drawn in one of the first
attempts to quantify the amount of air in ice, with the success not exceeding 50 – 66 %; there remained
33 – 50 % of air deemed to form bubbles too small to be discernible with the magnification of 160
times (Carte, 1961).
The present study suggests that the remaining solute is likely to be found in not only the puddles
of the highly concentrated solution but also the veins or grain boundary grooves threading the crystals.
The problem of the varying behavior of the surface brine puddles as opposed to the brine in the ice
grain grooves should be examined in more detail. The difference in the dynamics of these two
environments (Figure 6) suggests the possibility of physically and/or chemically distinct behaviors;
however, comprehensive research in the given respect is yet to be conducted.

### 4.2 Relevance to impurity locations anticipated in previous laboratory experiments

The observed strong dependence of the ice microstructure on the freezing method suggests that
inferring the locations of impurities exclusively from the way the sample was prepared is not
straightforward; subtle details, such as the particular sample geometry and material properties of its
cover, may play a significant role. Although we found satisfactory reproducibility of the freezing
methods, variability was noticed within our research and other investigations in cases when the
freezing rates had not been precisely controlled (Malley et al., 2018). Certain variance of values should
always be supposed, even when the cooling protocol is kept unaltered as the freezing remains
stochastic (Vetráková et al., 2017;Krausková et al., 2016;Heger et al., 2006;Heger and Klan, 2007).
Importantly, a generalizable conclusion can be drawn from our current observations: for instances
where the freezing occurs from the surface inwards, the impurities are to be expected inside the ice
matrix, whereas when the process starts from the cooled pad, brine expulsion to the ice surface is
likely to materialize. This justified and expectable result should be applied to review the experiments
in which the impurities' locations were inferred merely based on the freezing procedure.
Thus, we can also assume, exploiting current and previous observations (Krausko et al.,
2014;Blackford et al., 2007), that the impurities will be contained primarily inside ice spheres prepared
by solution nebulization into LN. Experimental procedures essentially identical with our method for
the preparation of the spheres were described by Kurkova et al. (2011); however, these authors
proposed that the impurities prevail on the outer surfaces of the ice spheres in the ice-air interface.
The difference between the present study and the approach adopted by Kurkova et al. lies in that salt
solutions are used in our case, whereas Kurkova et al. applied dilute organic compound solutions (with
$CuCl_2$ in some instances). We acknowledge that utilizing inorganic salts as a proxy for organic
compounds is only partly descriptive, but we assume that the compounds, inorganic or organic,
experience similar positions. It is interesting to notice that ice spheres prepared from a solution of 1,1-
diphenylethylene allowed complete conversion when exposed to ozone at 188 K but did not facilitate
the same at 268 K (Ray et al., 2013);  the lower temperatures possibly enabled the ozone to enter the
inside of the ice spheres, this being an ability  restricted by higher temperatures.   All our numerous
attempts to visualize low concentrations of organic compounds ended unsuccessfully, as we had not
observed significant differences from frozen pure water (data not shown).
In the present paper, we propose evidence for structural differences between the surface (Figures
2-7) and the interior of the investigated ice samples (Figure 9). In the ice broken via mechanical impact,
we cannot bring any evidence for the supposed cleavage "along defected sites such as veins and
pockets"(Kahan et al., 2010). From Fig. 9 we can infer that the sample disintegrated perpendicularly
to the lamellae. Certainly, after cutting the ice, some amount of impurities will surface; however, our
current observations indicate that most of the compounds still remain inside the ice interior. No
flowing out of the brine was observed at −25 °C, even though the sample had slowly sublimated.
Therefore, it seems unlikely that, by crushing the original ice cubes, the organic impurities from the
interior of the ice matrix could ascend to the ice-air interface in an appreciable quantity, as was
suggested by Kahan et al. (Kahan et al., 2010); such a process would probably not occur unless the
sample were exposed to increased temperatures for a prolonged time or strong sublimation.
Structurally, it can be proposed that even though the impurities frozen out of the solution may slightly
rearrange after the ice was crushed, they will not be present in the quasi-liquid layer but in the grooves
around the ice crystals. In our opinion, the faster disappearance in the experiments by Kahan and
Donaldson of the anthracene fluorescence signal in crushed ice as compared to ice cubes might rather
be explained by specific optical properties of crushed ice (McFall and Anastasio, 2016).

## 5   Conclusion

We documented high sensitivity of the brine-ice topology towards freezing procedures. The
experimental setup of our ESEM allowed us to image the brine puddles together with ice grain grooves
and free ice surfaces for a wide range of CsCl concentrations and various freezing methods previously
applied in the laboratory examination of ice. The ice grain sizes and the brine surface coverage were
found to be determined by not only the initial solution concentration but, crucially, also the freezing
method and the thermal history of the sample. The freezing method also influenced the sizes and
distribution of the air bubbles, which tend to co-locate with the brine. Therefore, the thermodynamic
state of the sample (the temperature and impurities' concentrations) is not the sole factor to
determine the shape of the frozen water: The freezing rate and directionality play a critical role in the
brine distribution and will thus influence the transport properties within sea ice, including the thermal
conductivity and permeability. The presented micrographs clarified the possible porosity and pore
microstructure of salty ices. Strong hysteresis resulting in the brine accumulating on the ice surface
was found during the warming-cooling cycle. The rheology of ice is strongly related to the freezing
rates; in our experiments, these were directly measured or inferred from the spacing of the lamellae
in the ice samples' interior, with the latter of these approaches being applied in cases of morphological
instability. The interior of the in-capsule, LN-frozen ice samples exhibited regularly spaced layers of ice
and brine and did not reconstruct during the observation. Besides the static snapshots, we were able
to document the ice sublimation and recrystallization dynamics. During the latter, the surface humps
retarded the ice grain boundary migration. The recognition of the impurities' locations within frozen
samples should help to reinterpret previous assumptions and to rationalize the observed behavior for
laboratory experiments, all with the goal to describe the complex interactions in natural ice. The
obtained knowledge can also serve as an input for physical and chemical models.

**Author contribution:** ĽV conducted the experiments and worked on the manuscript; JR and VN
conducted the experiments and constructed the microscope; DH proposed and supervised the
experiments and contributed to the writing of the manuscript. All of the authors contributed to the
discussions.

**Acknowledgements**: The research was supported by Czech Science Foundation (19-08239S). We thank Jiří Hudec for helping us with the thermal camera measurements and Vojtěch Svak for providing his know-how as regards the high-speed camera operation.

**Competing interests:** The authors declare no conflict of interest.

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
