# Peer review of "The Morphology of Ice and Liquid Brine in the Environmental SEM: A Study of the Freezing Methods"

_The Cryosphere, 2019_

## Referee Comment (RC1) · Anonymous Referee #1 · 18 Apr 2019

– General comments –

This paper presents a logical series of well-designed experiments which address a significant concern when studying the behavior of solute-containing ice samples prepared in a laboratory. Additionally, the results suggest additional experimental methods, and can be applied to behavior of natural systems in the cryosphere. The work is of high quality and is well presented, both in written form and visually. The use of Supplemental materials is particularly thorough and commendable. The work is publishable in its current form. However, I believe it can be improved either by clarifying or elaborating some specific points, listed below.

– Specific comments –

Line 113: The molar concentration of typical seawater is at the upper end of the experimental range of concentrations used here, and the authors do an excellent job discussing relevant concentration ranges in Line 583 ff. I believe it would aid the reader to have this concentration information presented earlier in the work, but this may be a matter of personal preference.

Lines 120-137: Did the authors have any suspicions or intentions on how each freezing method would cause the solutes to be organized in the sample? If so, it would be helpful to include that information here, even if (actually, especially if) those expectations were later found to be untrue.

Lines 138-141: While it makes sense to protect the surface from sublimating, how can the experimental procedure ensure the sublimation reveals only the original ice surface? Is it possible the ice surface is also sublimated to a certain degree, as subsequent experiments suggest? If so, what impact would this have?

Line 218: It is interesting that the surface became flooded with brine over the course of the procedure. How long did this take? Do you have a hypothesis what physical process would explain this?

Line 322-323: It may be useful to discuss the uncertainties this introduces into the analysis.

Line 315 ff: The idea of surface coverage is an interesting one. I think I understand the limitations of the technique, and I agree it is unfortunate the depth of the brine layer cannot be determined. However, is it worth including even some rough calculations of the total CsCl mass present in the sample, and determining if the freezing method assumptions are supportable? For example, you could calculate the surface area of the brine layer and assume it is present to a depth of 1,500 nm (the ESEM interrogation depth); does that volume account for all the CsCl, or just a tiny fraction of it?

As the authors (and some of the cited literature) suggest, assumptions about where freezing methods place solutes can be very speculative; any opportunity to constrain this information would be welcome.

Line 329: I'm not convinced the seeded sample would grow from the exposed surface downwards. Why wouldn't crystallization be favored at the basal surface, where presumably maximum heat transfer was happening and the droplet temperature was slightly lower?

Line 341-343: Was the droplet used here the same droplet presented in Figure 2b? If it is or if it isn't, I would clarify this point in the text. If it is a different droplet, what was the surface brine coverage for the droplet in 2b? This comparison might give the reader some idea of the variability in the system. Section 3.3.2 discusses surface coverage for the 0.05 M sample; while I understand it represents additional work, it may be worth at least tabulating surface coverage for all 9 samples presented in Figure 2 as a Supplemental figure.

Line 356-359: This, to me, is an interesting and important finding. As a suggestion, would it make sense to include the relative brine volume versus temperature (based on the phase diagram) as a second line on Figure 4, or add an additional scale on the right hand side indicating relative brine volume? This would allow a more direct comparison between brine surface area and brine volume.

Line 381-383: I'm a bit confused by the assertion that the brine volume has increased dramatically. You have previously stated the experimental method cannot evaluate brine volume, only surface area coverage. What data supports the idea that the brine volume has increased dramatically? Is it possible the brine volume is unchanged, and the brine has simply spread laterally? I would suggest addressing this uncertainty in the paper.

Line 432ff: While reading the first paragraph of this section, the question that came to mind was, "Would ice sublimation also explain the apparent motion of grain boundaries,

as the surface ice erodes?" a possibility I'm glad to see discussed in lines 486-490. I would suggest considering rewording some of the introductory statements of this section to acknowledge the motion is apparent, and may be an experimental artifact, not "true" motion. Do you have a sense how quickly sublimation occurs off the ice surface? The fact that the surface coverage of CsCl stays the same suggests sublimation is not happening on a timescale relevant for the apparent boundary movement, but an explicit reference to the amount of time required for sublimation (perhaps in section 3.4) would be welcome. It may be possible to combine the apparent lateral motion of the grain boundaries with a known sublimation rate to estimate the angle of the grain boundaries relative to the ice surface. If all the calculated angles are shallow, which would be geometrically unlikely, the finding would support the idea that the grain boundaries are actually moving and not just appearing to move.

Line 441: If it's easy to do, I would suggest adding text to the movie (Figure S1) including a timescale, or a caption stating the overall elapsed time in the movie. Otherwise the timescale is indirectly available in the caption for Figure 6.

Line 512: This line seems to suggest the method of freezing a droplet using a seed crystal should exclude impurities to the surface of the droplet during freezing. However, Line 329 suggests the opposite, that the droplet would crystallize from the surface downward. I suggest reconciling these two statements (or clarifying them if I have misunderstood either), or discussing it as an uncertainty.

Figures 744-749: This is a useful finding. It is worth noting that in at least one other related work, the same investigator suggested crushing an ice sample would preferentially expose impurities because the ice would cleave "along defect sites such as veins and pockets" (Kahan et al., 2010). If the research presented here can be used to address this issue as well, and it appears to me it could, I would suggest the text be modified accordingly.

– Technical corrections –

Line 266: Figure 21a should read Figure 2a. Line 373: Suggest that "expected" is a better word choice here than "expectable". Line 672: I think "special" should be replaced by "spatial". Line 729: I think that "expected" is a better word choice here than "expectable". Line 744: "Figure 2-7" should be "Figures 2-7". Line 769: I believe the correct temperature here should be -16 °C, not 16 °C.

– References –

Kahan, T. F., Zhao, R., Jumaa, K. B., and Donaldson, D. J.: Anthracene photolysis in aqueous solution and ice: Photon flux dependence and comparison of kinetics in bulk ice and at the air-ice interface, Environ Sci Technol, 44, 1302-1306, 10.1021/es9031612, 2010.

---

## Referee Comment (RC2) · Anonymous Referee #2 · 23 Apr 2019

This work describes the distribution of brine at the surface of frozen samples. It very convincingly presents experimental evidence from state-of-the-art SEM data for a distribution of the brine at two specific locations: grain boundaries and puddles. The impact of concentration and of freezing method are systematically discussed.

I support publication of the manuscript after a minor revision. The manuscript has a detailed, sound and easy to follow discussion and all conclusions are well supported by data. It is well written. The topic is of very high relevance and this work adds significantly to an ongoing discussion.

Main points 1. I'm a little confused about the detector sensitivity and variation in acqui-

[Figure]

Interactive
comment

sition settings. It peaks for the authors that they mention this point so explicitly. I agree with them, that this -while being nasty- does not impact their conclusions. Looking at Figure 3, this might imply that the samples with a high singal intensity, might even have a higer surface coverage of brine, because weaker spots from smaller puddles might be overseen. I might supect that sample g and f show the same fine structure as i their dark areas – but I would not think that this adds significantly to the brine covered area. However, in line 319 to 323 it is not clear to me how the "normlaised brine surface coverage" is defined and why this is free from this artefact. I would suggest to move the discussion from the experimental section here, use Figure 3 more explicitly to discuss the potential impact on your conclusions, and maybe even include estimates of this systematic error in the uncertainties mentioned in lines 315-320. 2. I suggest to add concentration and temperature information of the samples during measurement and ice production to the phase diagram (Figure 1). Throughout the text, I would encourage to be more precise in using the words brine volume, concentration, and salt amount. I'm not so convinced that you can discuss brine volume, as you are not sensitive to the thickness. Your observable is "only" the amount of salt, is it not? Would you expect the shock frozen samples (in N2) to crystalize once temperatures drop below the eutectic and could you comment on the question whether or not the salt deposits would liquify again upon warming to the temperature of investigation taken that this process might be slow. 3. Protecting the sample with a condensed ice layer is a excellent idea. However, could you give more details how you ensured that you "revealed the original surface" for the measurements and not removed too much ice (including some layers of the original surface). (lines 138)

Minor points

lines 12: I'm not a native speaker: What means "threading"? I find the first sentence of the abstract a slightly to abrupt and suggest to start more general.

lines 34: reword: ice and snow are also part of the environment.

Lines 41: Add reference to Grannas 2007 and/or Bartels-Rausch 2014

Lines 43: Rather "into the ice lattice at high concentration". Domine's work clearly shows that ice can hold impurities as solid solution and for some species at sufficient high concentration to impact partitioning to the atmosphere.

Lines 87: I found the discussion in this paragraph too detailed and slightly off topic and suggest to remove it. This paper does not add to the question of interior vs. surface distribution of impurities. It is only sensitive to the surface of the samples.

Lines 168: Is it correct that you never observed a supercooled solution? The brine always started crystalizing at the eutectic temperature? That would be an interesting observation.

Lines 171: What is an BSE detector?

Figure 2: Could you give an estimate on the smallest features that you would detect with the specific detector settings? (lines 213: no brine puddles larger than . . .. Were detected. . .).

In figure 4 the surface coverage seems to increase continioiusly, while T shows steps between -24 to -18°C. Could you comment on this?

---

## Referee Comment (RC3) · Sönke Maus (Referee) · 30 Apr 2019

**Review of manuscript tc-2019-13**
by Sönke Maus

This is a review of the manuscript *The Morphology of Ice and Liquid Brine in the Environmental SEM: A Study of the Freezing Methods* by L'ubica Vetrakova et al.

Below I cite from the Cryosphere Discussions manuscript tc-2019-13 in *italic font.*

**I  Summary**

The paper presents an analysis of images of ice samples with an environmental scanning electron microscope (SEM). All samples stem from laboratory freezing experiments of CsCl solutions, with different CsCl concentrations and different freezing conditions. The article is well written and structured into the sections 1.Introduction, 2. Methods, 3. Results and discussion, 4. Relevance to previous observations. Based on the results and discussion the authors conclude: *The findings thus quantify the amounts of brine exposed to incoming radiation, available for the gas exchange, and influencing other mechanical and optical properties of ice. The results have straightforward implications for artificially prepared and naturally occurring salty ices.*

I find the manuscript interesting, well written and to some degree worth publishing, as there is limited information available about the distribution of solute in ice formed in the environment, especially with the noted spatial resolution of 5 $\mu$m. However regarding the relevance for ice in the environment (section 4.) and the *straightforward implications for naturally occurring salty ices* proposed by the authors, I strongly disagree. While the paper puts many observations in the light of possible processes and phenomena described in the literature, the discussion is very vague. Some quantitative results are given but not compared to theories. It is neither clearly presented which of the observed ice-brine pattern are created by the method, and which relate to the freezing process. The following aspects illustrate my major concerns/ the weaknesses in the study:

I. While the authors apply different freezing methods, it is not clear to which degree these resemble natural freezing conditions. For (I), freezing of a filtered (0.45 $\mu$m) solution, in a silicon sample holder rather high supercoolings were likely to exist followed by unidirectional freezing. This may resemble nucleation in the atmosphere, yet filtering, silicon holder and unidirectionality introduce considerable differences. Method (III) and (IV), freezing within liquid nitrogene, is easy to perform but may lead to rather artificial freezing conditions. Hence only (II) seeding a slightly supercooled droplet with ice crystals resembles natural conditions. In general, the freezing of solutions is a morphological stability problem for which the pattern depends on temperature and solute gradients (that may be derived based on the degree of supercooling) and the solidification velocity. None of these boundary conditions is documented or monitored.

II. The technique only yields information about the distribution and pattern of ice and solute at the sample surface. For these brine films the thickness remains undetermined. The brine film coverage on the surface may not only depend on its (unknown) thickness, but also on processes in the interior of the samples. E.g. noting that brine will be expelled from the interior, the surface to volume ratio or specific surface area of samples

can be expected to be a highly relevant parameter. However, sample size has not been varied during the experiments. Another important property that affects expulsion of brine is the bulk density of an ice-brine aggregate. Some results obtained here (e.g., line 339-346) should be discussed in terms of the non-linear behaviour of the expansion coefficient of an ice brine-mixture that may take positive or negative values depending on solute concentration and temperature (e.g. Timco and Frederking, 1996). Overall, the discussion of brine coverage is incomplete and speculative - see specific notes below.

III. For all samples the author create (line 138) a thin layer of ice from condensed moisture on the samples surface, to protect the surface from sublimation. It should be tested how this thin layer of ice might affect the surface brine pattern: could the layer be porous and imply suction of brine into it? Does the creation of the layer change the surface temperature? May it lead to dissolution of ice in surface brine? How is the time determined at which the thin layer has sublimated? The behaviour of this layer needs to be described better, possibly through time-series of images showing its sublimation. In general, on the one hand there appear surface features that change during the imaging due to sublimation (grain boundaries), while others do not change (humps). On the other hand, the humps form during sublimation, while the grain boundaries have formed earlier, during freezing. It should be distinguished better which morphological parameters are forming during the freezing process and which during sample preparation and observation.

IV. Also the described dynamics of ice surfaces (3.5) appears to be related to sublimation during imaging and it is not clear for which process in natural ice the experiment with the present scales (time, sample size) is relevant. The description of these results and its discussion remain qualitative. Grain boundary migration in a temperature gradient is discussed, but no estimates of actual temperature gradients and migration velocities are given. It is also mentioned that tilting of grain boundaries with respect to the plane of sublimation might create an apparent rate of grain boundary migration, but no tests were made to verify this. The author's conclusions are vague: *We are, however, uncertain if the reason for these unusual dynamics of the frozen surface consists in one of the previously described conditions, both of them, or a completely different process.*

I would like to recommend the manuscript for publication, after these points have been adressed. The paper may also be improved by focusing on facts and validation of theories, rather than discussing a wide range of natural systems for which it does not provide new evident insight.

**II  Specific comments**

**1. Introduction**

L29 *straightforward implications* –> I would hardly call the results/conclusions as straightforward

L103–104 *and the location of sea water brine on sea ice can be inferred* –> sea ice is a rather different system, involving natural convection, unidirectional growth, a freeboard, the presence of snow, etc....

L138–141 –> This process of thin layer application and sublimation needs to be better shown/validated (VI. above)

**2. Methods**

L146– (2.2.) –> Here I would expect information about image resolution (mentioned later as 0.5 $\mu$m in the results section.

L163–168 *We estimate that the surface of a frozen sample is up to 2 ℃ warmer compared to that of the sample holder...*–> The crystallization temperature you observe is -25℃ which, allowing for some uncertainty, is within the eutectic temperature range you mention from other sources for aqueous CsCl solutions. There is thus no need to assume a warmer surface. Why is it assumed/likely then?

L170– (2.3.) –> The manual threshold segmentation process needs to be described better. What is the uncertainty in brine coverage related to segmentation uncertainty?

**3. Results and discussion**

L236– (table 1) –> These results are interesting. To understand their relevance for natural freezing processes they need however to be related to an estimate of freezing rates. Please consider model and other observation approaches to obtain such an estimate. Also, I suggest that for the freezing method IV, capsules in liquid nitrogen, the pattern shown in Figure 9 should be given here - see next note on L274–277. It would be further helpful to provide a table comparing the basic information for all methods (directionality, temperature difference, sample size, surface of observations).

L258–265 –> Crystal orientation is not retrieved by the imaging and its discussion is confusing. Supercooling and freezing rate alone explain different crystal sizes.

L274–277 *Therefore, based on analysing the crystal sizes, we can infer that the spontaneous freezing of the non-seeded droplet supercooled to about -16 ℃ occurred at the highest freezing rates experimentally attempted in this.*–> Based on the facets in Figure 9, I would rather suggest that freezing in LN2 with method IV was faster than in method I. Next, the freezing rate of droplets sprayed onto LN2 may have been be largest overall, with the difference that this interface has not become morphologically unstable. Such a planar

growth can be expected based on morphological stability theory for high growth rates (e.g., Sekerka, 1973; Coriell et al., 1994).

L309– (table 2) –> I would skip one decimal, if the image resolution is roughly 0.5 $\mu$mIt would be interesting to see statistics for higher temperature - this is discussed in detail for the brine surface coverage. Also, and estimate of salt content based on the surface grooves should be presented, to illustrate if pattern in the interior need to be different to reflect the nominal salt content.

L356-359 –>*Even though it is not possible to evaluate the volume from the microscopic images, we documented well that the brine surface coverage had risen four times during the warming, becoming much larger than the coverage implied by the phase diagram.*–> During cooling the coverage changed from 80 to 40%, consistent with the phase diagram. This indicates that thickness changes are sometimes involved and sometimes not. and should be discussed.

L379-381 –>*The results of this temperature cycling experiment indicate that, as the ice from the surface was required to melt during the heating in order to double the volume of the brine, a formerly inaccessible portion of the brine (restrained due to being trapped below the surface layer of the ice) surfaced.*–> The results can also be interpreted in terms of a thinning of the brine layer.

L395-397 –>*It cannot be excluded that the sample surface was several degrees warmer than the holder due to the effects of the electron beam; purging relatively warm gas through the specimen chamber.*–> Could you please present some numerical estimates?

L405-409 –>*Thus, the temperature increase with the subsequent spread of the brine on the surface may lead to the surface darkening, resulting in higher solar radiation absorption and further increase of the temperature.*–> The effect of a brine film on sea ice albedo has to my knowledge not been discussed yet - rather the number and size of inclusions in the bulk ice are relevant.

L432–434 (and whole 3.5.) –>*The dynamics of the frozen samples' surfaces could be observed even at -23 ℃ . The CsCl brine-filled grain boundaries were not static: their positions changed swiftly in time. Interestingly, the positions of the humps on the ice surface did not change accordingly.* –> The two explanations (i) migration of grain boundaries in a temperature gradient and (ii) exposure of inclined grain boundaries during surface sublimation are not so expected and their discussion is lengthy. Could they be validated by some data and estimates (possible temperature gradients, sublimation rates, internal sample observations, etc...)?.

L526-528 –> 10 $\mu$m indicates rather fast freezing - see note on freezing rates above.

**4. Relevance to previous observations**

L578–591 –> It is corrrect that the range 0.005 to 0.5M solute concentration is found in nature. However, the classification into relevant salinity regimes and processes in nature

sounds a bit artificial: sea ice can have 0.05 M solute content (brackish ice, Baltic Sea ice), and for salt concentration near roads I would expect a large range depending on environmental conditions. Surface snow on sea ice may be much more saline than 0.005 M, and a range of 0.01 to 0.1 M is more representative.

L592-595 –> *The principal finding presented within our study is embodied in the very strong sensitivity of the ice-brine morphology and brine distribution to the freezing method (Figure 2, 3). Even for identical solution concentrations, the method and direction of freezing strongly modify the appearance of the ice surface morphology.*–> These principal findings are not new.

L601-610 –> It is not clear that natural convection played a role for solute redistribution within the different freezing experiments, as freezing was either fast or upwards. The comparison to sea ice formation and desalination is thus not useful here. While directionality is important, it is not due to natural convection.

L662–664 –>*Apparently, the amount of the brine on the ice surface was not sufficient to fill the groove around all the ice grains due to low brine concentration and/or the freezing method concentrating the brine towards the interior of the ice matrix*–> As mentioned, directionality, local brine expulsion and bulk ice-brine density changes are relevant here and this statement needs to be validated.

L678 –>*frozen aqueous solution (without added salt)*–> Do you mean frozen (almost pure) water?.

L710–712 –>*The present study suggests that the remaining solute is likely to be found in not only the puddles of the highly concentrated solution but also, the veins or grain boundary grooves threading the crystals.*–> This is not a new observation. To produce something new you should at least give some estimates of the solute contained in the veins/grain boundaries, based on your observations.

**5. Conclusion**

L759 –>*amount of brine*–> You have not determined the amount, just the surface coverage.

L766–767–>*The presented micrographs clarified the possible porosity and pore microstructure of sea ice.*–> The relevance of the freezing conditions/sample sizes for sea ice has not been demonstrated. Regarding microstructure of sea ice there are textbooks that 'clarify' this much better (e.g., Weeks, 2010; Shokr and Sinha, 2015).

L768–770–>*From the ice crystal sizes we infer the actual 769 freezing rates*–> In fact no freezing rates are inferred. Also, freezing rates are very likely largest for the LN2 freezing (III and IV) - see discussion above.

**References**

Coriell, S. R., Boisvert, R. F., MacFadden, G. B., Brush, L. N., Favier, J. J., 1994. Morphological stability of a binary alloy during directional soldification: initial transient. J. Cryst. Growth 140 (1-2), 139–147.

Sekerka, R. F., 1973. Morphological stability. North-Holland Publ. Co, Ch. 15, pp. 403–441.

Shokr, M., Sinha, N., 2015. Sea ice: physics and remote sensing. Geophysical Monograph 209. John Wilkey and Sons.

Timco, G., Frederking, R., 1996. A review of sea ice density. Cold Regions Science Technol. 24, 1–6.

Weeks, W. F., 2010. On Sea Ice. University of Alaska Press.

---

## Author Comment (AC1) · 6 Jun 2019

We thank the referee for their helpful, constructive comments, which enabled us to reconsider our interpretations, discuss some details more clearly, and eliminate relevant problems.

The red sections comprise the comments proposed by Reviewer 1; our responses are outlined in black; and the modified portions of the manuscript are shown in italics.

– Specific comments –
Line 113: The molar concentration of typical seawater is at the upper end of the experimental range of concentrations used here, and the authors do an excellent job discussing relevant concentration ranges in Line 583 ff. I believe it would aid the reader to have this concentration information presented earlier in the work, but this may be a matter of personal preference.

We have added to this section the information about the natural occurrence of salt in the snow and ice, as shown below:

*The concentrations are proxies for natural salt occurrence in coastal snow (5 mM) (Beine et al., 2012;Douglas et al., 2012), up to the concentration in sea water (0.5 M) (Massom et al., 2001;Thomas, 2017).*

Lines 120-137: Did the authors have any suspicions or intentions on how each freezing method would cause the solutes to be organized in the sample? If so, it would be helpful to include that information here, even if (actually, especially if) those expectations were later found to be untrue.

Thank you for the comment; our preference is to leave the descriptive part without personal opinions. The relevant expectations are outlined in the Discussion section to a certain extent.

*As mentioned above, the ice crystal size difference between the samples frozen under high and low supercooling was expected. However, we were rather surprised that the ice crystals in the non-seeded supercooled droplet appeared smaller than those in the ice spheres prepared by spraying a solution into the LN.*

Lines 138-141: While it makes sense to protect the surface from sublimating, how can the experimental procedure ensure the sublimation reveals only the original ice surface? Is it possible the ice surface is also sublimated to a certain degree, as subsequent experiments suggest? If so, what impact would this have?

The responses are provided in a related part of the paper, as shown below:

*The layer exhibited a structure very different from that the frozen samples (Figure S18); thus, the desublimed ice was always readily distinguishable from the original ice sample. The sublimation of the condensed layer was monitored at the start of the ESEM imaging; as soon as the layer had sublimed, the imaged surface of the frozen sample was affected by the sublimation process to only a very small extent. We can infer this fact because we had also followed further sublimation of the ice samples, as will be reported in the near future. By further extension, the effect of ice sublimation could be more pronounced in temperature cycling and ice dynamics experiments.*

Line 218: It is interesting that the surface became flooded with brine over the course of the procedure. How long did this take? Do you have a hypothesis what physical process would explain this?

The event most likely resulted from the slow, gradual sublimation of the ice. The flooding typically took about 5 minutes in the given samples.

*during the procedure, the surface became completely brine-flooded due to the slow, gradual ice sublimation.*

Line 322-323: It may be useful to discuss the uncertainties this introduces into the analysis.

As this comment was proposed also by Reviewer 2 we are giving same answer.

The uncertainty of the surface coverage calculation associated with the manual threshold selection can be indirectly deduced from the variance of the surface coverage in the sequence of the images recorded at the same temperature in Figure 4. Five images were recorded at the temperature -23.4 °C at the beginning of the experiment; their estimated surface coverages equaled 22.5, 21.1, 22.9, 20.4, and 25.8 %. Similarly, the values of 36.0, 35.7, and 34.4 were estimated for the surface coverage of the sample at -25.1 °C. Thus, we assume the surface coverage uncertainty due to the manual threshold selection is in units of percent. The systematic error due to a difference in the signal intensity would be larger, and therefore only the surface coverages based on the samples with similar signal intensities can be directly compared.

We modified and extended the Ch. "2.3 Estimating the brine surface coverage" accordingly:

*The uncertainty of the manual threshold selection procedure is indirectly deducible from the variance of the surface coverage in the sequence of the images recorded at the same temperature soon after one another; the relevant value was in units of percent.*

Line 315 ff: The idea of surface coverage is an interesting one. I think I understand the limitations of the technique, and I agree it is unfortunate the depth of the brine layer cannot be determined. However, is it worth including even some rough calculations of the total CsCl mass present in the sample, and determining if the freezing method assumptions are supportable? For example, you could calculate the surface area of the brine layer and assume it is present to a depth of 1,500 nm (the ESEM interrogation depth); does that volume account for all the CsCl, or just a tiny fraction of it?

The details of the performed calculations have been added to the Supplementary information section and theirs results into the text:

*we can nevertheless estimate the relative amount of the salt on the surface of the sample, presuming that the thickness of the surface brine is similar to channels' width. Therefore, we estimated the relative amount of the salt on the surface of the frozen samples prepared from 0.05 M CsCl via the freezing methods I-III, exploiting the assumption that the thickness of the surface layer equals 1.5 µm. As a result, the relative amounts of the salt on the surface were 9, 4.5, and 26% for the non-seeded droplets, seeded droplets, and ice spheres, respectively. The calculations are provided in the SI.*

As the authors (and some of the cited literature) suggest, assumptions about where freezing methods place solutes can be very speculative; any opportunity to constrain this information would be welcome.

We consider the current observation a suitable contribution to the general discussion on these issues; further experimental methods are being designed to facilitate better understanding of the problem.

Line 329: I'm not convinced the seeded sample would grow from the exposed surface downwards. Why wouldn't crystallization be favored at the basal surface, where presumably maximum heat transfer was happening and the droplet temperature was slightly lower?

The temperature of the sample was only 2 °C below the freezing point of pure water (and slightly less in an aqueous solution), and it remained constant within the sample. Spontaneous crystalization did not occur; this process started after the sample had been seeded with an ice crystal positioned onto the sample's surface. Horizontal movement of the ice-solution interface could be observed, proceeding from the point of touch between the sample and the ice crystal towards the other side of the sample. Although the vertical movement of the interface could not be monitored, we assume that the crystallization had started at the surface, where the crystallization nucleus was in contact with the droplet. The temperature gradient was supposed to be minimal, as the solution had been thermostated before seeding at -2°C. The thermal camera measurement suggested thermal stabilization within 3 minutes at -18 °C, and it will probably be less for -2°C. Furthermore, a dedicated study proposes that crystalization on the surface is 1e10 more likely than in the bulk (Shaw et al., 2005).

We modified the text accordingly:

*Conversely, the seeded sample probably started to crystallize from the surface, close to the edge of the sample as the first place in touch with the seeding crystals. Even though small vertical temperature gradients were to be expected through the solution, we considered crystallization from the surface more likely, in accordance with a related study which had suggested that crystallization on the surface was $1×10^{10}$ more probable than the corresponding process in the bulk (Shaw et al., 2005).*

Line 341-343: Was the droplet used here the same droplet presented in Figure 2b? If it is or if it isn't, I would clarify this point in the text. If it is a different droplet, what was the surface brine coverage for the droplet in 2b? This comparison might give the reader some idea of the variability in the system. Section 3.3.2 discusses surface coverage for the 0.05 M sample; while I understand it represents additional work, it may be worth at least tabulating surface coverage for all 9 samples presented in Figure 2 as a Supplemental figure.

The applied droplet was the same as that presented in Figure 2b; the observed spot, however, was different. The surface coverage variabilities are summarized in Table S3, where the coverage for 5 independent, identically prepared samples is presented in relation to each freezing method. We prefer avoiding a direct comparison of the surface coverage values for various concentrations, as this may become misleading due to the arguments presented in Section 2.3.

Line 356-359: This, to me, is an interesting and important finding. As a suggestion, would it make sense to include the relative brine volume versus temperature (based on the phase diagram) as a second line on Figure 4, or add an additional scale on the right-hand side indicating relative brine volume? This would allow a more direct comparison between brine surface area and brine volume.

We are grateful for good idea. Figure 4 and the text has been completed with the relevant information.

*The strong relationship between the brine volume and surface coverage is indirectly demonstrated in Figure 4; the coverage follows the same trend as the relative amount of brine in the frozen sample ($w_{brine}$) calculated from the phase diagram using the formula*

$$w_{brine} = \frac{m_{aq}}{m_{fr} - m_{aq}},$$

*where, $m_{aq}$ is the molality of the aqueous solution of salt (that can be approximated by the concentration of the salt in case of low concentrations) and $m_{fr}$ is the molality of the brine in the frozen sample (it is dependent on temperature and can be inferred from the liquidus curve of the phase diagram). Thus, we are of the opinion that the change of the brine volume is the main cause of the surface coverage variation.*

Line 381-383: I'm a bit confused by the assertion that the brine volume has increased dramatically. You have previously stated the experimental method cannot evaluate brine volume, only surface area coverage. What data supports the idea that the brine volume has increased dramatically? Is it possible the brine volume is unchanged, and the brine has simply spread laterally? I would suggest addressing this uncertainty in the paper.

We cannot evaluate the brine volume directly; such a procedure is possible only in the surface coverage, and therefore the corresponding formulation in the text may have been rather vague. Nevertheless, we suppose the brine volume and the surface coverage are related parameters, as spreading of the brine on the surface of the ice is governed mainly by the surface tension.

The following text has been incorporated in the manuscript:

*Even though it is not possible to evaluate the volume from the microscopic images, we documented well that the brine surface coverage had risen four times during the warming, becoming much larger than the increase implied by the phase diagram. Presumably, the brine volume can be inferred indirectly from the surface coverage if spreading of the brine on the ice surface is governed mainly by the surface tension. We interpret the change in the brine volume as the central cause of the observed surface coverage variations, and we also suppose that the observed 4-fold rise in the coverage indicates an even larger volume alteration because the brine thickness will increase too. The strong relationship between the brine volume and surface coverage is indirectly demonstrated in Figure 4;*

Line 432ff: While reading the first paragraph of this section, the question that came to mind was, "Would ice sublimation also explain the apparent motion of grain boundaries, as the surface ice erodes?" a possibility I'm glad to see discussed in lines 486-490. I would suggest considering rewording some of the introductory statements of this section to acknowledge the motion is apparent, and may be an experimental artifact, not "true" motion.

Thank you! We appreciate the suggestions. The appearance of the ice boundaries' movement is discussed from the beginning of the paragraph now.

*With the time progressing, the grain boundary on the right-hand side performed an apparent gradual movement through the hump, while the hump did not change.*

Do you have a sense how quickly sublimation occurs off the ice surface? The fact that the surface coverage of CsCl stays the same suggests sublimation is not happening on a timescale relevant for the apparent boundary movement, but an explicit reference to the amount of time required for sublimation (perhaps in section 3.4) would be welcome. It may be possible to combine the apparent lateral motion of the grain boundaries with a known sublimation rate to estimate the angle of the grain boundaries relative to the ice surface. If all the calculated angles are shallow, which would be geometrically unlikely, the finding would support the idea that the grain boundaries are actually moving and not just appearing to move.

We materialized the answer in the text of the manuscript:

*We do not know the complete time required for our samples to sublimate, because the process was not aimed at in our experiments; contrariwise, we intended to prevent sublimation of the samples. We usually imaged the samples for about 30 minutes, and they never sublimated during this period. The sample height at the start of the imaging corresponded to approximately 1 mm (based on the working distance of the ESEM). The upper limit of the (vertical) sublimation rate can thus be calculated if full sublimation of the sample is assumed to materialize within 30 minutes; the (vertical) sublimation rate would then be 33 µm/min. The brine channel (Figure 6) moved by ⬚19 µm/min. If the apparent movement of the brine channel is caused solely by the sublimation, the angle between the channel and the horizontal line will be 60°.*

However, the uncertainty of the sublimation rate discourages us from discussing the further details in the manuscript.

Line 441: If it's easy to do, I would suggest adding text to the movie (Figure S1) including a timescale, or a caption stating the overall elapsed time in the movie. Otherwise the timescale is indirectly available in the caption for Figure 6.

The video has been completed with the elapsed time.

Line 512: This line seems to suggest the method of freezing a droplet using a seed crystal should exclude impurities to the surface of the droplet during freezing. However, Line 329 suggests the opposite, that the droplet would crystallize from the surface downward. I suggest reconciling these two statements (or clarifying them if I have misunderstood either), or discussing it as an uncertainty.

Thank you for identifying the conflicting statements. We attempted recording the process on camera; however, this did not deepen our knowledge in this respect thus far. We are of the opinion that, as discussed above, the freezing proceeds from the surface laterally and also vertically downwards. The fact that we did not observe that many air bubbles may have been caused by freezing slow enough to allow the bubbles to escape, either to the ambient air or towards the bottom of the sample.

*Conversely, the seeding method allows slow ice crystalization, leading to major expulsion of the air either out of the solution or downward, towards the bottom; we are nevertheless unsure of the details of such scenarios.*

Figures 744-749: This is a useful finding. It is worth noting that in at least one other related work, the same investigator suggested crushing an ice sample would preferentially expose impurities because the ice would cleave "along defect sites such as veins and pockets" (Kahan et al., 2010). If the research presented here can be used to address this issue as well, and it appears to me it could, I would suggest the text be modified accordingly.

Thank you for identifying the connection. We have changed the text accordingly.

*In the present paper, we propose evidence for structural differences between the surface (Figures 2-7) and the interior of the investigated ice samples (Figure 9). In the ice broken via mechanical impact, we cannot bring any evidence for the supposed cleavage "along defected sites such as veins and pockets"(Kahan et al., 2010). From Fig. 9 we can infer that the sample disintegrated perpendicularly to the lamellae. Certainly, after cutting the ice, some amount of impurities will surface; however, our*

*current observations indicate that most of the compounds still remain inside the ice interior. No flowing out of the brine was observed at −25 °C, even though the sample had slowly sublimated.*

– Technical corrections –

Line 266: Figure 21a should read Figure 2a. Line 373: Suggest that "expected" is a better word choice here than "expectable". Line 672: I think "special" should be replaced by "spatial". Line 729: I think that "expected" is a better word choice here than "expectable". Line 744: "Figure 2-7" should be "Figures 2-7".

Thank you for careful reading – good suggestions.

Line 769: I believe the correct temperature here should be -16 °C, not 16 °C.

We think that 16-degrees supercooling refers to -16 °C.

– References –

Kahan, T. F., Zhao, R., Jumaa, K. B., and Donaldson, D. J.: Anthracene photolysis in aqueous solution and ice: Photon flux dependence and comparison of kinetics in bulk ice and at the air-ice interface, Environ Sci Technol, 44, 1302-1306, 10.1021/es9031612, 2010.

Beine, H., Anastasio, C., Domine, F., Douglas, T., Barret, M., France, J., King, M., Hall, S., and Ullmann, K.: Soluble chromophores in marine snow, seawater, sea ice and frost flowers near Barrow, Alaska, Journal of Geophysical Research, 117, 10.1029/2011jd016650, 2012.

Douglas, T. A., Domine, F., Barret, M., Anastasio, C., Beine, H. J., Bottenheim, J., Grannas, A., Houdier, S., Netcheva, S., Rowland, G., Staebler, R., and Steffen, A.: Frost flowers growing in the Arctic ocean-atmosphere–sea ice–snow interface: 1. Chemical composition, Journal of Geophysical Research, 117, 10.1029/2011jd016460, 2012.

Massom, R. A., Eicken, H., Hass, C., Jeffries, M. O., Drinkwater, M. R., Sturm, M., Worby, A. P., Wu, X., Lytle, V. I., Ushio, S., Morris, K., Reid, P. A., Warren, S. G., and Allison, I.: Snow on Antarctic sea ice, Rev. Geophys., 39, 413-445, 10.1029/2000rg000085, 2001.

Shaw, R. A., Durant, A. J., and Mi, Y.: Heterogeneous Surface Crystallization Observed in Undercooled Water, The Journal of Physical Chemistry B, 109, 9865-9868, 10.1021/jp0506336, 2005.

Thomas, D. N.: Sea Ice, Wiley-Blackwell, Chichester, UK, 2017.

---

## Author Comment (AC2) · 7 Jun 2019

We thank the referee for their helpful, constructive comments, which enabled us to reconsider our interpretations, discuss some details more clearly, and eliminate relevant problems.

The red sections comprise the comments proposed by Reviewer 2; our responses are outlined in black; *and the modified portions of the manuscript are shown in italics.*

Anonymous Referee #2

Main points

1. I'm a little confused about the detector sensitivity and variation in acquisition settings. It peaks for the authors that they mention this point so explicitly. I agree with them, that this -while being nasty- does not impact their conclusions. Looking at Figure 3, this might imply that the samples with a high signal intensity, might even have a higher surface coverage of brine, because weaker spots from smaller puddles might be overseen. I might suspect that sample g and f show the same fine structure as i their dark areas – but I would not think that this adds significantly to the brine covered area. However, in line 319 to 323 it is not clear to me how the "normalised brine surface coverage" is defined and why this is free from this artefact. I would suggest to move the discussion from the experimental section here, use Figure 3 more explicitly to discuss the potential impact on your conclusions, and maybe even include estimates of this systematic error in the uncertainties mentioned in lines 315-320.

The formulation "the samples with a high signal intensity might even have a higher surface coverage of brine because weaker spots from smaller puddles might be overseen " is true.
       The uncertainty of the surface coverage calculation associated with the manual threshold selection can be indirectly deduced from the variance of the surface coverage in the sequence of the images recorded at the same temperature in Figure 4.  Five images were recorded at the temperature -23.4 °C at the beginning of the experiment; their estimated surface coverages equaled 22.5, 21.1, 22.9, 20.4, and 25.8 %. Similarly, the values of 36.0, 35.7, and 34.4 were estimated for the surface coverage of the sample at -25.1 °C. Thus, we assume the surface coverage uncertainty due to the manual threshold selection is in units of percent. The systematic error due to a difference in the signal intensity would be larger, and therefore only the surface coverages based on the samples with similar signal intensities can be directly compared.

We modified and extended the Ch. "2.3 Estimating the brine surface coverage" accordingly:

*The uncertainty of the manual threshold selection procedure is indirectly deducible from the variance of the surface coverage in the sequence of the images recorded at the same temperature soon after one another; the relevant value was in units of percent.*

2. I suggest to add concentration and temperature information of the samples during measurement and ice production to the phase diagram (Figure 1).

As the displayed phase diagram ranges from 0 to 18 mol kg$^{-1}$, the two lower salt concentrations (0.005 M and 0.05 M) would be on the very left of the phase diagram, thus becoming hardly recognizable.

Throughout the text, I would encourage to be more precise in using the words brine volume, concentration, and salt amount. I'm not so convinced that you can discuss brine volume, as you are not sensitive to the thickness. Your observable is "only" the amount of salt, is it not?

Thank you for the comment.
What we observed was the surface coverage with the salt. However, we suppose that the brine volume and surface coverage are related parameters, as spreading of the brine on the surface of the ice is governed mainly by the surface tension. The relationship between the brine coverage and volume can be inferred from Figure 4. We attempted to use the indicated words with greater care throughout the text.

Would you expect the shock frozen samples (in N2) to crystalize once temperatures drop below the eutectic and could you comment on the question whether or not the salt deposits would liquify again upon warming to the temperature of investigation taken that this process might be slow.

We know from the differential scanning calorimetry (related to the NaCl solution, paper accepted for publication in J. Chemical Physics: "Vitrification and Increase of Basicity in between Ice Ih Crystals in Rapidly Frozen Dilute NaCl Aqueous Solutions") that the brine in the shock-frozen samples crystallizes at some temperature below $T_{eu}$, depending on the cooling rate. In our ESEM micrograms, crystalized CsCl was not observed in the frozen spheres due to poor thermal conductivity between the cooling stage and the spheres; however, we examined CsCl crystalization in the freezing method IV, where the thermal conductivity was sufficient. The deposited salt liquefies upon warming; Figure S19 was added to demonstrate the process.

3. Protecting the sample with a condensed ice layer is a excellent idea. However, could you give more details how you ensured that you "revealed the original surface" for the measurements and not removed too much ice (including some layers of the original surface). (lines 138)

The formation of an ice layer on the surface of the frozen sample cannot be excluded unless the sample has been frozen in an inert atmosphere. At the start of the imaging, we identified a condensed ice layer on the surface of the sample. The surface's structure is very different from that of the actual samples, as demonstrated in Figure S18. We imaged the samples from the very beginning until the ice sublimated completely. The data were evaluated by using the images recorded soon after the experiment had begun; thus, the samples' surfaces are not at all or only very slightly affected by the sublimation.

*The layer exhibited a structure very different from that the frozen samples (Figure S18); thus, the desublimed ice was always readily distinguishable from the original ice sample. The sublimation of the condensed layer was monitored at the start of the ESEM imaging; as soon as the layer had sublimed, the imaged surface of the frozen sample was affected by the sublimation process to only a very small extent. We can infer this fact because we had also followed further sublimation of the ice samples, as will be reported in the near future. By further extension, the effect of ice sublimation could be more pronounced in temperature cycling and ice dynamics experiments.*

Minor points
lines 12: I'm not a native speaker: What means "threading"? I find the first sentence of the abstract a slightly to abrupt and suggest to start more general.

The word "threading" was adopted from (Cheng et al., 2010) in the hope of its being an acceptable option. In the context of the article, *threading* means *winding around*: (https://www.dictionary.com/browse/threading?s=t)

lines 34: reword: ice and snow are also part of the environment.

Certainly: The problem has been eliminated.

Lines 41: Add reference to Grannas 2007 and/or Bartels-Rausch 2014
The reference has been added.

Lines 43: Rather "into the ice lattice at high concentration". Domine's work clearly shows that ice can hold impurities as solid solution and for some species at sufficient high concentration to impact partitioning to the atmosphere.

The relevant text section has been expanded accordingly.

*Excepting low concentrations of HF, $NH_4^+$, HCl, $HNO_3$ and formaldehyde (Perrier et al., 2002;Thibert and Domine, 1998, 1997), impurities are usually not incorporated into the ice lattice (Krausková et al., 2016;Hobbs, 2010;Wilson and Haymet, 2008).*

Lines 87: I found the discussion in this paragraph too detailed and slightly off topic and suggest to remove it. This paper does not add to the question of interior vs. surface distribution of impurities. It is only sensitive to the surface of the samples.

The interior of the frozen samples was, in fact, approached by using the method 4, where the samples frozen in the test tubes had been broken and examined. Another applicable technique of reaching inside the sample was sublimation.

Lines 168: Is it correct that you never observed a supercooled solution? The brine always started crystalizing at the eutectic temperature? That would be an interesting observation.

Thank you for the comment. Regrettably, we cannot discuss your point properly, as we did not measure the temperature of the sample in a direct manner; the temperature of the cooling stage was adjusted. The temperature difference between the sample and the cooling stage is characterized in the Methods section. Moreover, the cooling is markedly slower in the sample than in the Peltier stage. We nevertheless know that supercooling is common in the DSC of NaCl solutions.

Lines 171: What is an BSE detector?

The abbreviation refers to a detector of back-scattered electrons.

Figure 2: Could you give an estimate on the smallest features that you would detect with the specific detector settings? (lines 213: no brine puddles larger than.... Were detected...).

We imaged the samples with the resolution of 0.5 μm. However, the size of the structures is not the decisive factor determining the visibility of the structures. The detector settings are influenced by the salt concentration and surface roughness. Further, if the crystalline salt is present, the visibility of other structures becomes greatly reduced because the salt signal is intensive too, and the detector sensitivity is thus reduced.

In figure 4 the surface coverage seems to increase continioiusly, while T shows steps between -24 to -18°C. Could you comment on this?

The temperature in Figure 4 is that set on the Peltier stage. When the temperature of the stage has been changed, it takes some time for the temperature of the sample to settle; thus, the changes on the surface coverage are continuous, not stepwise.

Hobbs, P. V.: Ice Physics, OUP Oxford, 2010.

Cheng, J., Soetjipto, C., Hoffmann, M. R., and Colussi, A. J.: Confocal Fluorescence Microscopy of the Morphology and Composition of Interstitial Fluids in Freezing Electrolyte Solutions, Journal of Physical Chemistry Letters, 1, 374-378, 10.1021/jz9000888, 2010.

Krausková, Ľ., Procházková, J., Klašková, M., Filipová, L., Chaloupková, R., Malý, S., Damborský, J., and Heger, D.: Suppression of protein inactivation during freezing by minimizing pH changes using ionic cryoprotectants, International Journal of Pharmaceutics, 509, 41-49, 10.1016/j.ijpharm.2016.05.031, 2016.

Perrier, S., Houdier, S., Domine, F., Cabanes, A., Legagneux, L., Sumner, A. L., and Shepson, P. B.: Formaldehyde in Arctic snow. Incorporation into ice particles and evolution in the snowpack, Atmos. Environ., 36, 2695-2705, 2002.

Thibert, E., and Domine, F.: Thermodynamics and kinetics of the solid solution of HCl in ice, J. Phys. Chem. B, 101, 3554-3565, 1997.

Thibert, E., and Domine, F.: Thermodynamics and kinetics of the solid solution of HNO3 in ice, J. Phys. Chem. B, 102, 4432-4439, 1998.

Wilson, P. W., and Haymet, A. D. J.: Workman-Reynolds freezing potential measurements between ice and dilute salt solutions for single ice crystal faces, J. Phys. Chem. B, 112, 11750-11755, 2008.

---

## Author Comment (AC3) · 7 Jun 2019

We thank Dr. Soenke Maus for an extensive and detailed review, which, in our opinion, has enabled us to substantially improve the manuscript.

The red sections comprise the comments proposed by Reviewer 1; our responses are outlined in black; *and the modified portions of the manuscript are shown in italics.*

I find the manuscript interesting, well written and to some degree worth publishing, as there is limited information available about the distribution of solute in ice formed in the environment, especially with the noted spatial resolution of 5 _m. However regarding the relevance for ice in the environment (section 4.) and the straightforward implications for naturally occurring salty ices proposed by the authors, I strongly disagree.

We agree that the "*straightforwardness*" of the presented results is directed more towards laboratory-prepared samples than natural ices. As we work exclusively with the former, we may not have sufficiently appreciated the specific requirements of the latter in the original manuscript. After realizing this, we modified the text accordingly.

While the paper puts many observations in the light of possible processes and phenomena described in the literature, the discussion is very vague. Some quantitative results are given but not compared to theories.

We definitely agree in that the article misses the approach of a theoretician and would like to incorporate such an aspect in the future; currently, however, we focus on experimental microscopy, which, previously, was not covered in the literature on a comprehensive basis.

It is neither clearly presented which of the observed ice-brine pattern are created by the method, and which relate to the freezing process.

The text has been revised to better separate and highlight these facts.

I. While the authors apply different freezing methods, it is not clear to which degree these resemble natural freezing conditions. For (I), freezing of a filtered (0.45 _m) solution, in a silicon sample holder rather high supercoolings were likely to exist followed by unidirectional freezing. This may resemble nucleation in the atmosphere, yet filtering, silicon holder and unidirectionality introduce considerable differences. Method (III) and (IV), freezing within liquid nitrogene, is easy to perform but may lead to rather artificial freezing conditions. Hence only (II) seeding a slightly supercooled droplet with ice crystals resembles natural conditions.

We appreciate the expert analysis of the relevance towards natural freezing conditions. As a matter of fact, our experiments are indeed centered especially on laboratory-frozen samples, and the step towards conditions closer to environmental freezing is yet to be taken.

In general, the freezing of solutions is a morphological stability problem for which the pattern depends on temperature and solute gradients (that may be derived based on the degree of supercooling) and the solidification velocity. None of these boundary conditions is documented or monitored.

We have made an attempt to determine the freezing velocity, when technically possible.

II. The technique only yields information about the distribution and pattern of ice and solute at the sample surface. For these brine films the thickness remains undetermined. The brine film coverage on the surface may not only depend on its (unknown) thickness, but also on processes in the interior of the samples. E.g. noting that brine will be expelled from the interior, the surface to volume ratio or specific surface area of samples can be expected to be a highly relevant parameter. However, sample size has not been varied during the experiments. Another important property that affects expulsion of brine is the bulk density of an ice-brine aggregate. Some results obtained here (e.g., line 339-346) should be discussed in terms of the non-linear behaviour of the expansion coefficient of an ice brine-mixture that may take positive or negative values depending on solute concentration and temperature (e.g. Timco and Frederking, 1996). Overall, the discussion of brine coverage is incomplete and speculative - see specific notes below.

III. For all samples the author create (line 138) a thin layer of ice from condensed moisture on the samples surface, to protect the surface from sublimation. It should be tested how this thin layer of ice might affect the surface brine pattern:

A deeper description of the desublimed layer has been added, together with the relevant images (Figure S18).

could the layer be porous and imply suction of brine into it?

The condensed layer of ice is porous; we do not assume the layer could alter the underlaying ice surface in a notable manner. The sample is exposed to the condensed layer only briefly, for the period of a few minutes.

Does the creation of the layer change the surface temperature?

The formation of the condensed layer was inevitable in the open-chamber freezing. The layer was formed gradually during cooling the frozen sample; thus, no abrupt temperature change was observed. During lowering the sample temperature, the surface of the sample was likely warmer than the bottom due to imperfect heating; the condensed layer could also participate in slower cooling of the surface. The experiments, however, were conducted after the sample temperature had been thermally equilibrated and the condensed layer had sublimated.

May it lead to dissolution of ice in surface brine? How is the time determined at which the thin layer has sublimated? The behaviour of this layer needs to be described better, possibly through time-series of images showing its sublimation.

The images showing the sublimation of the condensed layer were incorporated in the SI (Figure S18). In our experiments, the time of the presence of the desublimated ice layer was kept short, and we thus do not expect the brine to wick in substantially. Other details on the layer sublimation are given below.

In general, on the one hand there appear surface features that change during the imaging due to sublimation (grain boundaries), while others do not change (humps). On the other hand, the humps form during sublimation, while the grain boundaries have formed earlier, during freezing. It should be distinguished better which morphological parameters are forming during the freezing process and which during sample preparation and observation.

We have modified the text to distinguish the effects of the freezing from those caused by the sample sublimation.

IV. Also the described dynamics of ice surfaces (3.5) appears to be related to sublimation during imaging and it is not clear for which process in natural ice the experiment with the present scales (time, sample size) is relevant. The description of these results and its discussion remain qualitative. Grain boundary migration in a temperature gradient is discussed, but no estimates of actual temperature gradients and migration velocities are given. It is also mentioned that tilting of grain boundaries with respect to the plane of sublimation might create an apparent rate of grain boundary migration, but no tests were made to verify this. The author's conclusions are vague: We are, however, uncertain if the reason for these unusual dynamics of the frozen surface consists in one of the previously described conditions, both of them, or a completely different process.

We admit that the low pressure sample sublimation is only an approximation to the natural conditions, progressing much faster than would have been relevant to sea ice surface or frozen aerosols. The approximation may be relevant to a certain extent if the surface reconstruction occurs at a low rate; such an assumption nevertheless remains to be validated.

**1. Introduction**
L29 straightforward implications –> I would hardly call the results/conclusions as straightforward

The portion of the text has been modified to read as follows:

*The results have straightforward and indirect implications for artificially prepared and naturally occurring salty ices, respectively.*

L103–104 and the location of sea water brine on sea ice can be inferred –> sea ice is a rather different system, involving natural convection, unidirectional growth, a freeboard, the presence of snow, etc....

The sentence has been modified to read as follows:

*thus, the presence of brine on the surface of the frozen samples can be clearly monitored, with indirect implications towards the location of sea water brine on sea ice.*

L138–141 –> This process of thin layer application and sublimation needs to be better

The formation of an ice layer on the surface of the frozen sample cannot be excluded unless the sample has been frozen in an inert atmosphere. At the start of the imaging, we identified a condensed ice layer on the surface of the sample. The surface's structure is very different from that of the actual samples, as demonstrated in Figure S18. We imaged the samples from the very beginning until the ice sublimated completely. The data were evaluated by using the images recorded soon after the experiment had begun; thus, the samples' surfaces are not at all or only very slightly affected by the sublimation.

A discussion of these details has been incorporated in the manuscript, as follows:

*The layer exhibited a structure very different from that the frozen samples (Figure S18); thus, the desublimed ice was always readily distinguishable from the original ice sample. The sublimation of the condensed layer was monitored at the start of the ESEM imaging; as soon as the layer had sublimed, the imaged surface of the frozen sample was affected by the sublimation process to only a very small extent. We can infer this fact because we had also followed further sublimation of the ice samples, as will be reported in the near future. By further extension, the effect of ice sublimation could be more pronounced in temperature cycling and ice dynamics experiments.*

**2. Methods**
L146– (2.2.) –> Here I would expect information about image resolution (mentioned later as 0.5 μm in the results section.

A sentence concerning the resolution has been added to complete the methods:

*The images were typically recorded with the magnification of 500 (image resolution of ∼ 0.5 μm), although a resolution up to ten times higher is feasible.*

L163–168 We estimate that the surface of a frozen sample is up to 2 °C warmer compared to that of the sample holder...–> The crystallization temperature you observe is -25 °C which, allowing for some uncertainty, is within the eutectic temperature range you mention from other sources for aqueous CsCl solutions. There is thus no need to assume a warmer surface. Why is it assumed/likely then?

Our suspicion of a warmer surface was based on the observed CsCl crystalization. Based on the referee's comment, however, we employed a thermal camera to study the temperature behavior more closely. The details are now specified within the manuscript:

*The thermal camera measurements in atmospheric conditions (without the reduced pressure environment of the ESEM sample chamber) showed that the sample was about 2 °C warmer than the cooling stage. Further warming of the sample during the ESEM experiments was to be expected, especially due to the effects of the electron beam and the relatively warm gas purged through the specimen chamber; conversely, the surface cooling embodied an expectable process too, considering the ice sublimation and water evaporation from the brine. These factors could contribute to the salt crystallization on the sample surface, even when the temperature of the holder does not change; still,*

*the overall temperature difference between the surface of the frozen samples and the cooling stage will likely not exceed 2 °C.*

L170– (2.3.) –> The manual threshold segmentation process needs to be described better. What is the uncertainty in brine coverage related to segmentation uncertainty?

In this context, we propose the following explanation, that is also detailed in Chapter 2.3:

The pixels of the grayscale image recorded with the ESEM exhibit brightness in the range from 0 (black) to 255 (white). The Mountain ® Software enables us to select pixels with the brightness value above the chosen threshold; these pixels are highlighted in the software. By overlaying the highlighted mask and the white areas in the original image, the best-fitting threshold is chosen to represent the brine and to exclude the ice (Figure S1).

The uncertainty of the manual threshold selection procedure is indirectly deducible from the variance of the surface coverage in the sequence of the images recorded at the same temperature, Figure 4. Five images were recorded at the temperature -23.4 °C at the beginning of the experiment; their estimated surface coverages equalled 22.5, 21.1, 22.9, 20.4, and 25.8 %. Similarly, the values of 36.0, 35.7, and 34.4 were estimated for the surface coverage of the sample at -25.1 °C. Thus, we assume that the surface coverage uncertainty due to the manual threshold selection ranged within units of percent.

**3. Results and discussion**

L236– (table 1) –> These results are interesting. To understand their relevance for natural freezing processes they need however to be related to an estimate of freezing rates. Please consider model and other observation approaches to obtain such an estimate. Also, I suggest that for the freezing method IV, capsules in liquid nitrogen, the pattern shown in Figure 9 should be given here - see next note on L274–277. It would be further helpful to provide a f comparing the basic information for all methods (directionality, temperature difference, sample size, surface of observations).

We performed additional open-chamber freezing experiments, utilizing a high-speed camera to determine the freezing rate in the methods I and II. The movement of the freezing front was recorded for the 0.05 M CsCl solution. In the freezing method I, the freezing rate of 150 mm/s was established; in the technique II, the freezing rate of 0.2 mm/s was roughly estimated, as the freezing interface was very poorly visible on the camera in this freezing process. Regrettably, the camera could not be used to monitor the freezing front in the methods III and IV (freezing in liquid nitrogen).

As regards the structure of the text, we would like to separate the observation of the surfaces from that of the samples' interiors, thus facilitating separation of the samples frozen with the method IV.

A table comparing all the freezing methods has been incorporated in the text. (Table 1)

L258–265 –> Crystal orientation is not retrieved by the imaging and its discussion is confusing. Supercooling and freezing rate alone explain different crystal sizes.

We agree: The text concerned has been deleted.

L274–277 Therefore, based on analysing the crystal sizes, we can infer that the spontaneous freezing of the non-seeded droplet supercooled to about -16 °C occurred at the highest freezing rates experimentally attempted in this.–> Based on the facets in Figure 9, I would rather suggest that freezing in LN2 with method IV was faster than in method I. Next, the freezing rate of droplets sprayed onto LN2 may have been be largest overall, with the difference that this interface has not become morphologically unstable. Such a planar growth can be expected based on morphological stability theory for high growth rates (e.g., Sekerka, 1973; Coriell et al., 1994).

We thank the reviewer for referring to the literature and appreciate his valuable comment. The text has been modified accordingly.

*An alternative reason for comparatively large ice crystals forming during the nebulization of a solution into LN may consist in a different freezing mechanism: at freezing rates from 210 to 2×10$^{-6}$ mm/s, morphological instability of the ice surface is predicted, whereas outside this range linear progression of the ice front is expected (Wettlaufer, 1992;Maus, 2019).*

The method IV allows the estimation of freezing rate to be less than 0.1 mm/s.

*Using the theoretically predicted instability for the 0.035 % NaCl solution, (Wettlaufer, 1992) we can conclude, for the observed 10-$\mu$m spacing, that the freezing rate can range from 0.1 to 10$^{-5}$ mm/s. Thus, the freezing of the 16 °C supercooled sample can proceed 1,000 times (or more) faster than that of the LN-immersed capsules.*

L309– (table 2) –> I would skip one decimal, if the image resolution is roughly 0.5 µm. It would be interesting to see statistics for higher temperature - this is discussed in detail for the brine surface coverage. Also, and estimate of salt content based on the surface grooves should be presented, to illustrate if pattern in the interior need to be different to reflect the nominal salt content.

One decimal place was omitted in the Table. The grain boundary groove widths are difficult to measure at higher temperatures, as a large portion of the ice surface is covered with brine pools. The surface salt content has been estimated, and the calculations are provided in the SI.

L356-359 –>Even though it is not possible to evaluate the volume from the microscopic images, we documented well that the brine surface coverage had risen four times during the warming, becoming much larger than the coverage implied by the phase diagram.–> During cooling the coverage changed from 80 to 40%, consistent with the phase diagram. This indicates that thickness changes are sometimes involved and sometimes not. And should be discussed.

We do not have any reliable method to determine the thickness of the brine layer; however, we suppose the area of the layer is, to some extent, related to the brine volume (see below).

L379-381 –>The results of this temperature cycling experiment indicate that, as the ice from the surface was required to melt during the heating in order to double the volume of the brine, a formerly inaccessible portion of the brine (restrained due to being trapped below the surface layer of the ice) surfaced.–> The results can also be interpreted in terms of a thinning of the brine layer.

It cannot be excluded that the brine layer became thinner and more widespread. However, we suppose that the ratios of the volume and thickness of the layer are related, as spreading of the brine on the surface is governed mainly by the surface tension. In Figure 4, the surface coverage is compared to the amount of brine in the sample, calculated from the phase diagram; it clearly shows that the two parameters are related.

The following text has been added:
*Presumably, the brine volume can be inferred indirectly from the surface coverage if spreading of the brine on the ice surface is governed mainly by the surface tension. We interpret the change in the brine volume as the central cause of the observed surface coverage variations, and we also suppose that the observed 4-fold rise in the coverage indicates an even larger volume alteration because the brine thickness will increase too.*

L395-397 –>It cannot be excluded that the sample surface was several degrees warmer than the holder due to the effects of the electron beam; purging relatively warm gas through the specimen chamber.–> Could you please present some numerical estimates?

As already cited above the text has been rephrased as follows:
*The thermal camera measurements in atmospheric conditions (without the reduced pressure environment of the ESEM sample chamber) showed that the sample was about 2 °C warmer than the cooling stage. Further warming of the sample during the ESEM experiments was to be expected, especially due to the effects of the electron beam and the relatively warm gas purged through the specimen chamber; conversely, the surface cooling embodied an expectable process too, considering the ice sublimation and water evaporation from the brine. These factors could contribute to the salt crystallization on the sample surface, even when the temperature of the holder does not change; still, the overall temperature difference between the surface of the frozen samples and the cooling stage will likely not exceed 2 °C.*

L405-409 –>Thus, the temperature increase with the subsequent spread of the brine on the surface may lead to the surface darkening, resulting in higher solar radiation absorption and further increase of the temperature.–> The effect of a brine film on sea ice albedo has to my knowledge not been discussed yet - rather the number and size of inclusions in the bulk ice are relevant.

The number and sizes of inclusions are certainly factors of considerable importance.(Light et al., 2003) However, the optical properties are undoubtedly influenced also by the phase state of the salt/brine, as measured and discussed in (Light et al., 2016;Carns et al., 2016) The specific references have been added together with an explanatory sentence:

*Conversely, the brine crystallizing on the ice surface would result in larger reflection of the radiation, as the crystals would scatter the light substantially more effectively compared to the liquid brine (Carns et al., 2016;Light et al., 2016). These effects, together with the number and sizes of inclusions, ice grains, and location and characteristics of the absorbing particles, influence the overall absorption properties of icy bodies (Warren, 2019).*

L432–434 (and whole 3.5.) –>The dynamics of the frozen samples' surfaces could be observed even at -23 °C. The CsCl brine-filled grain boundaries were not static: their positions changed swiftly in time. Interestingly, the positions of the humps on the ice surface did not change accordingly. –> The two explanations (i) migration of grain boundaries in a temperature gradient and (ii) exposure of inclined grain boundaries during surface sublimation are not so expected and their discussion is lengthy. Could they be validated by some data and estimates (possible temperature gradients, sublimation rates, internal sample observations, etc...)?.

We incorporated the answer for both explanation into the text:
(i)

*The sample was prepared from a hemispherical droplet by cooling from the bottom. Thus, the height was most likely not constant across the sample and would be the most prominent in the center, allowing the temperature gradient to rise across the surface. A thermal camera was used to estimate the temperature gradients throughout the sample during cooling outside the ESEM chamber; it was not experimentally feasible to insert the camera into the chamber. As the temperature of the sample was dropping from 0 to −18 °C, the temperature at the center of the surface was approximately 2.7 °C higher than at the periphery. This difference had been caused presumably by the greater thickness of the sample at the central section. However, after the sample was cooled down and its temperature became approximately constant (which typically took about 3 minutes), the gradient was reduced, albeit in an opposite manner: The center was about 0.2-0.3 °C colder than the periphery, producing a small temperature gradient throughout the sample even after the cooling.*

*The sample presented in Figure 6 was sufficiently cooled down, as enough time had elapsed since the last temperature adjustment; therefore, we assume there might be a temperature gradient of only several tenths of °C across the sample.*

(ii)

*The sublimation rate largely depends on both the amount of water vapor purged into the chamber and the temperature. We do not know the complete time required for our samples to sublimate, because the process was not aimed at in our experiments; contrariwise, we intended to prevent sublimation of the samples. We usually imaged the samples for about 30 minutes, and they never sublimated during this period. The sample height at the start of the imaging corresponded to approximately 1 mm (based on the working distance of the ESEM). The upper limit of the (vertical) sublimation rate can thus be calculated if full sublimation of the sample is assumed to materialize within 30 minutes; the (vertical) sublimation rate would then be 33 µm/min. The brine channel (Figure 6) moved by ⊡19 µm/min. If the apparent movement of the brine channel is caused solely by the sublimation, the angle between the channel and the horizontal line will be 60°.*

L526-528 –> 10 µm indicates rather fast freezing - see note on freezing rates above.

We very much appreciate the help that Dr. Soenke Maus provided through interpreting our observations. We have added the following text:

*Using the theoretically predicted instability for the 0.035 % NaCl solution, (Wettlaufer, 1992) we can conclude, for the observed 10-$\mu$m spacing, that the freezing rate can range from 0.1 to $10^{-5}$ mm/s. Thus, the freezing of the 16 °C supercooled sample can proceed 1,000 times (or more) faster than that of the LN-immersed capsules.*

**4. Relevance to previous observations**

L578–591 –> It is corrrect that the range 0.005 to 0.5M solute concentration is found in nature. However, the classification into relevant salinity regimes and processes in nature sounds a bit artificial: sea ice can have 0.05 M solute content (brackish ice, Baltic Sea ice), and for salt concentration near roads I would expect a large range depending on environmental conditions. Surface snow on sea ice may be much more saline than 0.005 M, and a range of 0.01 to 0.1 M is more representative.

We appreciate the expert comment and advice by Dr. Soenke Maus; we have changed the text accordingly:

*The tested concentrations ranged over two orders of magnitude. The values within 500 - 50 mM define the concentration of NaCl in seawater, and therefore they are also descriptive of fresh sea ice in the given context (Massom et al., 2001;Thomas, 2017). NaCl concentrations reaching up to 160 mM were detected immediately next to a highway treated against road icing; 50 mM of a salt solution can thus be considered a concentration potentially found farther from roads or also in their close vicinity when the salt was already partly flushed away (Notz and Worster, 2009;Labadia and Buttle, 1996). The values discovered in surface snows in Arctic coastal regions or on the surface of frost flowers span between 10-100 mM but may also be as low as 5 mM.(Beine et al., 2012;Douglas et al., 2012;Maus, 2019).*

L592-595 –>The principal finding presented within our study is embodied in the very strong sensitivity of the ice-brine morphology and brine distribution to the freezing method (Figure 2, 3). Even for identical solution concentrations, the method and direction of freezing strongly modify the appearance of the ice surface morphology.–> These principal findings are not new.

Yes, we agree on that point, as relevant items of literature and also theoretical models have been composed and released. With our experiments, we primarily intended to understand the locations of impurities in laboratory-made ices.

L601-610 –> It is not clear that natural convection played a role for solute redistribution within the different freezing experiments, as freezing was either fast or upwards. The comparison to sea ice formation and desalination is thus not useful here. While directionality is important, it is not due to natural convection.

The comparison to natural convection indeed is rather remote from the central subject; our original intention was only to summon the convection from that field. The following text has been complemented in relation to this information:

*Our freezing methods differ from that related to natural sea water in the rates and directionality. By extension, we would like to imply that the role of material convection in relation to the drop size should be considered for the modeling of the freezing process.*

L662–664 –>Apparently, the amount of the brine on the ice surface was not sufficient
to fill the groove around all the ice grains due to low brine concentration and/or the freezing method concentrating the brine towards the interior of the ice matrix–> As mentioned, directionality, local brine expulsion and bulk ice-brine density changes are relevant here and this statement needs to be validated.

We agree with the comment and have attempted to discuss the topic accordingly. However, we are unsure of further validation besides the presented micrograms.

L678 –>frozen aqueous solution (without added salt)–> Do you mean frozen (almost pure) water?.

Yes, we meant pure water. Thank you!

L710–712 –>The present study suggests that the remaining solute is likely to be found in not only the puddles of the highly concentrated solution but also, the veins or grain boundary grooves threading the crystals.–> This is not a new observation. To produce something new you should at least give some estimates of the solute contained in the veins/grain boundaries, based on your observations.

Estimates of the relative amount of the solute present on the surface layer have been added to the text. The calculations are outlined in the SI.

*As a result, the relative amounts of the salt on the surface were 9, 4.5, and 26% for the non-seeded droplets, seeded droplets, and ice spheres, respectively.*

**5. Conclusion**
L759 –>amount of brine–> You have not determined the amount, just the surface coverage.

We agree: "surface coverage" is a more accurate expression.

L766–767–>The presented micrographs clarified the possible porosity and pore microstructure
of sea ice.–> The relevance of the freezing conditions/sample sizes for sea ice has not
been demonstrated. Regarding microstructure of sea ice there are textbooks that 'clarify'
this much better (e.g., Weeks, 2010; Shokr and Sinha, 2015).

We agree: Our sample preparation methods are not directly related to sea ice formation. The sentence has been modified as follows:

*The presented micrographs clarified the possible porosity and pore microstructure of salty ices.*
L768–770–>From the ice crystal sizes we infer the actual 769 freezing rates–> In fact
no freezing rates are inferred. Also, freezing rates are very likely largest for the LN2
freezing (III and IV) - see discussion above.

If our interpretation of the lamellar spacing in the samples prepared via the method IV is correct, the freezing rate could be comparable to or slower than those in the method II. Nevertheless, in the Conclusion we abandoned the inference of the freezing rate from the grain sizes, as it does not seem very reliable after we have studied Dr. Soenke Maus' comments and explanations.

Instead we state:

*The rheology of ice is strongly related to the freezing rates; in our experiments, these were directly measured or inferred from the spacing of the lamellae in the ice samples' interior, with the latter of these approaches being applied in cases of morphological instability.*